# Simulating multi-hazard event sets for life cycle consequence analysis

Leandro Iannacone[1], Kenneth Otárola[2], Roberto Gentile[1], and Carmine Galasso[1]

[1]University College London, UK
[2]Scuola Universitaria Superiore Pavia (IUSS), Italy

**Correspondence:** Leandro Iannacone (l.iannacone@ucl.ac.uk)

**Abstract.** In the context of natural hazard risk quantification and modeling of hazard interactions, some literature separates "Level I" (or occurrence) interactions from "Level II" (or consequence) interactions. The Level I interactions occur inherently due to the nature of the hazards, independently of the presence of physical assets. In such cases, one hazard event triggers or modifies the occurrence of another (e.g., heavy rain and flooding; liquefaction and landslides triggered by an earthquake), thus creating a dependency between the features characterizing such hazard events. They differ from Level II interactions, which instead occur through impacts/consequences on physical assets/components and systems (e.g., accumulation of physical damage or social impacts due to earthquake sequences, landslides due to the earthquake-induced collapse of a retaining structure). Multi-hazard Life Cycle Consequence (LCCon) analysis aims to quantify the consequences (e.g., repair costs, downtime, casualty rates) throughout a system's service life and should account for both Level I and II interactions. The available literature generally considers Level I interactions – the focus of this study – mainly defining relevant taxonomies, often qualitatively, without providing a computational framework to simulate a sequence of hazard events incorporating the identified interrelations among them. This paper addresses this gap, proposing modeling approaches associated with different types of Level I interactions. It describes a simulation-based method for generating multi-hazard event sets (i.e., a sequence of hazard events and associated features throughout the system's life cycle) based on the theory of competing Poisson processes. The proposed approach incorporates the different types of interactions in a sequential Monte Carlo sampling method. The method outputs multi-hazard event sets that can be integrated into LCCon frameworks to quantify interacting hazard consequences. An application incorporating several hazard interactions is presented to illustrate the potential of the proposed method.

## 1 Introduction

The modeling and quantification of catastrophe risk throughout a system's service life must encompass the occurrence of multiple natural and anthropogenic hazards. In fact, the occurrence of multiple events within a short time span (whether dictated by a causality between events or by sheer coincidence) may subject the system to exacerbated economic and societal consequences (e.g., de Ruiter et al., 2020). Such consequences have been increasing over the past decades (e.g., Di Baldassarre et al., 2018) due to several factors such as climate change, urbanization, and globalization (e.g., Cutter et al., 2008; Cutter, 2018). The assessment of such consequences cannot be approached by merely overlaying methodologies for individual hazard types. In the

context of this paper, a (natural) hazard type refers to a specific category of natural events or conditions that have the potential to cause harm or damage. Indeed, multi-hazard risk analysis must account for the interactions among various hazard types/events and their corresponding impacts/consequences. Efforts have been devoted to standardizing the nomenclature for multi-hazard risk analysis (e.g., Kappes et al., 2012; Gardoni and LaFave, 2016) to enhance effective communication among end users and various stakeholders. In general, the literature refers to compound hazards (and risks) whenever there is an interaction of physical phenomena/processes (i.e., natural-hazard events and their consequences) across multiple spatial and temporal scales (e.g., Pescaroli and Alexander, 2018; Sadegh et al., 2018; Zscheischler et al., 2018). However, existing research also acknowledges the importance of separating the interactions among compound hazards into *occurrence interactions*, which do not depend on assets/components and systems (including social ones) affected by hazard events, and *impact/consequence interactions* that can only occur through these exposed elements. Zaghi et al. (2016) classified the former as *Level I interactions* and the latter as *Level II interactions*. Level I interactions are due to dependencies in hazard frequencies/characteristics or the triggering or intensifying effect of one hazard type upon another. Gill and Malamud (2014) extensively reviewed Level I interactions, categorizing them based on the physical correlations between their occurrences and examining several hazard types to identify those capable of triggering or amplifying others. Nevertheless, the works mentioned above primarily categorize interactions qualitatively; they lack a discussion on the computational tools required to integrate these interactions into simulation-based frameworks for risk modeling and quantification. Consequently, the challenge of simulating sequences of events that incorporate the identified interactions largely remains unexplored. Some studies have tried to address this task. Still, they either have limited scope (e.g., site-specific and scenario-based studies like Adachi and Ellingwood, 2008; and Marzocchi et al., 2012) or treat all interaction types uniformly, irrespective of their distinct characteristics (e.g., Mignan et al., 2014). The challenges associated with obtaining realistic sequences of events have led multiple authors to select specific, representative scenarios in their multi-hazard assessments, disregarding the Level I interactions in favor of a detailed study of Level II interactions (e.g., Nofal et al., 2023).

In this paper, we present a simulation-based methodology to generate sequences of hazard events (in terms of their time of occurrence and features), denoted as *event sets*, throughout the lifespan of a system (typically spanning 50 to 100 years, depending on the system's socio-economic significance). Our approach incorporates the various types of Level I interactions found in the literature, each specified by appropriate modeling techniques. We distinguish between concurrent interactions (i.e., when hazards coincide in time and space) and successive interactions (i.e., when a primary hazard precedes a secondary one). Moreover, within successive interactions, we distinguish between those where a primary hazard immediately triggers secondary events (i.e., triggering interactions) and those where a primary hazard alters the occurrence rate of secondary hazard types (i.e., altering interactions). The proposed simulation-based method assumes hazard events can be modeled as competing Poisson point processes. Non-homogeneous Poisson processes can be incorporated by transforming them into equivalent homogeneous processes (e.g., Westcott, 1977) or leveraging thinning methods for interarrival time simulation (e.g., Lewis and Shedler, 1979), as demonstrated in this paper. The outcome is a sequential Monte Carlo (MC) approach enabling efficient simulation of multi-hazard event sets. These event sets can then be integrated into frameworks for Level II interactions (e.g., Selva, 2013; Dehghani

et al., 2021; Otárola et al., 2023a, b), facilitating the quantification of consequences for the purposes of Life Cycle Consequence (LCCon) Analysis.

The rest of the paper is organized as follows: Section 2 lists the types of hazard classifications and which dimension/information should be considered when generating multi-hazard event sets. Section 3 presents the proposed methodology. Section 4 shows how the procedure can be applied in practice with realistic data for an idealized case-study location. Finally, Section 5 summarizes the paper's content and proposes ideas for future work on the topic.

## 2 Hazard classifications

Hazard types can be classified based on various dimensions, influencing how they are modeled in a multi-hazard context. This paper specifically focuses on classifications directly influencing the simulation of multi-hazard event sets (i.e., a sequence of hazard events and associated features throughout the system's life cycle) at a given target location, i.e., where there is exposure in the form of "people, infrastructure, housing, production capacities, and other tangible human assets" (UNISDR, 2005). Additional classifications could be identified based on the spatial extent, spatial variability, and spatial dependence of hazard types (e.g., Gill and Malamud, 2017). However, such classifications are outside the scope of this paper. We note that spatial considerations can be integrated into the models applied within the proposed framework (e.g., when establishing the distance between the location of interest and the location of the simulated hazard occurrence). Yet, for a more comprehensive approach to multi-hazard event sets simulation at a larger scale (e.g., at a regional scale), one could explicitly account for various types of spatial correlations (e.g., in terms of hazard characteristics or local intensities, as defined below).

The proposed methodology employs the exceedance rates associated with different severity measures of the hazard events (see Section 3 for a detailed definition of a severity measure) to simulate the interarrival times between events. The proposed algorithm is agnostic toward the specific physical factors that govern the numerical values of such rates (which can be obtained from physics-based or empirical models). Thus, we do not consider classifications like those presented in Shaluf (2007), which separate natural from man-made hazard types. We also exclude the hazard type classifications outlined in the literature review by Gill and Malamud (2014, 2017) that refer to the specific causes of the hazard events, such as hydrological, atmospheric, or geophysical factors.

As a result, the developed algorithm incorporates the following three dimensions in the simulation of a multi-hazard event set (Figure 1):

– **Dependency**: Hazard types could be classified as *independent* if their occurrence/severity is not affected by the concurrent or preceding occurrence/severity of other hazard events or *dependent* if their occurrence/severity can be attributed to the occurrence/severity of other hazard events. In the case of dependent hazard types/events, the proposed methodology accounts for the types of interactions identified in the literature, namely concurrent and successive interactions (e.g., Zaghi et al., 2016). *Concurrent interactions* between two or more hazard types can be identified whenever the hazard types/events tend to occur simultaneously and/or to overlap for a period of time (e.g., storm surge, waves, and strong wind that co-occur during a hurricane). In the case of *successive interactions*, instead, a causal relationship exists between a

primary hazard type/event and one or more secondary hazard types/events. According to these causal relationships, two broad categories can be identified within successive interactions. We denote *triggering* the interactions where the secondary hazard type/event (or multiple secondary hazard types/events) is triggered immediately after the occurrence of the primary hazard type (e.g., liquefaction immediately following an earthquake). In contrast, *altering* interactions are those where the rate of occurrence of the secondary hazard type (or multiple secondary hazard types) increases (or, more generally, changes) following the occurrence of the primary hazard type (e.g., aftershocks following a mainshock). The resulting classification of interactions is a combination of the qualitative classifications proposed by Zaghi et al. (2016) (concurrent vs. successive) and Gill and Malamud (2014) (interactions where a hazard event is triggered vs. interactions where the probability of a hazard event is increased). Section 3 describes how such classifications affect the numerical modeling of multi-hazard event sets.

– **Duration**: Hazard types can be grouped into two categories based on their duration. Specifically, following the classification of disasters proposed by the Sendai Framework for Disaster Risk Reduction 2015-2030 (UNISDR, 2005), we separate sudden-onset hazard types from slow-onset hazard types. *Sudden-onset* hazard types are characterized by a sudden and brief occurrence that can be modeled as a single point in time (e.g., earthquakes). *Slow-onset* hazard types, instead, have a detectable start and end point (e.g., pandemics, droughts) and occur over an extended period.

– **Temporal variability**: Hazard types can be classified as time-independent or time-dependent based on their rate of occurrence over time. The rate of *time-independent* hazard types is constant over time (as such, the occurrence of these types of hazards is typically modeled with homogeneous processes), while the rate of *time-dependent* hazard types varies due to physical factors that affect their probability of occurrence within a given time window (as such, the occurrence of these types of hazards is typically modeled with non-homogeneous processes). For example, it is reasonable to assume time independence for mainshocks generating from large seismogenic zones incorporating multiple faults with similar rupture rates/characteristics (e.g., Der Kiureghian and Ang, 1977). As such, in these cases, the occurrence of mainshocks is modeled with homogeneous Poisson processes (e.g., Abrahamson and Bommer, 2005). On the other hand, aftershocks are typically modeled with non-homogeneous processes as their rate of occurrence typically reduces as a function of the time elapsed from the occurrence of the mainshock (e.g., Utsu, 1970). Hazard types could fall into different categories based on the complexity of the considered occurrence models. Advanced seismic hazard modeling approaches, for instance, may also consider the time-varying modeling of mainshock rates (e.g., Anagnos and Kiremidjian, 1988; Iacoletti et al., 2022), while simplified models for aftershocks may use a constant rate of occurrence that produces on average the same number of events as the non-homogeneous process in a given time window (e.g., Iervolino et al., 2014; Iervolino and Giorgio, 2022). Time-dependent hazard types can be further separated into *seasonal* if their rate of occurrence periodically changes over time (e.g., the occurrence of heavy rains modeled as a function of the season), *increasing* if their rate of occurrence increases over time (e.g., heavy rains under climate change effects) and *decaying*, if their rate of occurrence decreases over time (e.g., aftershocks).

Figure 1 summarizes the classifications relevant to the algorithm proposed in Section 3. These classifications are orthogonal, i.e., they categorize hazard types based on independent criteria that do not overlap with each other. Each hazard type can be assigned to three categories based on its dependency, duration, and temporal variability.

    As the primary goal of the proposed methodology is to seamlessly incorporate the available classifications for hazard inter-actions within a mathematical framework for hazard event simulation, the next section provides the modeling implications of

130 the highlighted classifications.

## 3   Methodology

Each hazard type is associated with certain *event characteristics* (e.g., rupture characteristics and magnitude in the case of earthquakes, rainfall characteristics - intensity and duration - in the case of extreme rainfall events). These quantities only characterize an event and do not account for local effects at a system's or site's location (e.g., local soil properties of a specific

site, distance from the earthquake source, topography of the area). We denote the curves relating the event characteristics to their corresponding exceedance rates as *event curves* (e.g., magnitude-frequency distributions in seismic hazard analysis or intensity-duration-frequency in extreme rainfall event analysis). Through appropriate modeling, the event characteristics are typically translated into site- and/or system-specific *intensity measures* (e.g., ground shaking for structures with selected vibration periods, flood depths), which are typically used as an input to obtain the corresponding physical impacts caused by

the event at a given location (e.g., through fragility/vulnerability models, e.g., Gentile et al., 2022). Appropriate methods for the local intensity calculation can be found in the literature and depend on the hazard type considered. For example, Ground Motion Models (GMMs) (e.g., Douglas and Edwards, 2016) can be used to translate earthquake characteristics into earthquake-induced ground-motion intensity measures such as peak ground parameters (i.e., peak ground acceleration, velocity, and displacement) and pseudo-spectral accelerations, among others. Such models could also account for the spatial and cross-intensity correlation

of the intensity measure (e.g., Jayaram and Baker, 2010a, b). For floods, accurate flow-based hydraulic models can be used to translate the rainfall characteristics into flood depths at different locations (e.g., Mignot and Dewals, 2022). In general, analyses at the regional scale call for maps displaying the spatial variability of the intensity measure across different locations. Finally, the methods to generate intensity measures can be integrated within end-to-end probabilistic frameworks to obtain curves linking each intensity measure value to its associated exceedance rate in a given time window (i.e., one year in case of

annual exceedance rates). Such curves are denoted as *hazard curves* (e.g., the curves for the exceedance of a given wind speed and surge depth in Apivatanagul et al., 2011, or the hazard curves from the Global Earthquake Model Seismic Hazard Map, Pagani et al., 2020). Because both event and hazard curves are used interchangeably in the proposed formulation for the same purpose on a case-by-case basis, we arbitrarily introduce the term *rate curves* to refer collectively to both cases and the term *severity measure* to refer to both event characteristics and intensity measures. In fact, some aspects of multi-hazard event sets

may be governed by the event characteristics (i.e., the rate of aftershocks is governed by the magnitude of the mainshock). In contrast, others may be governed by the intensity measures (i.e., the triggering of a landslide following the occurrence of an earthquake is governed by the ground motion at the slope location).

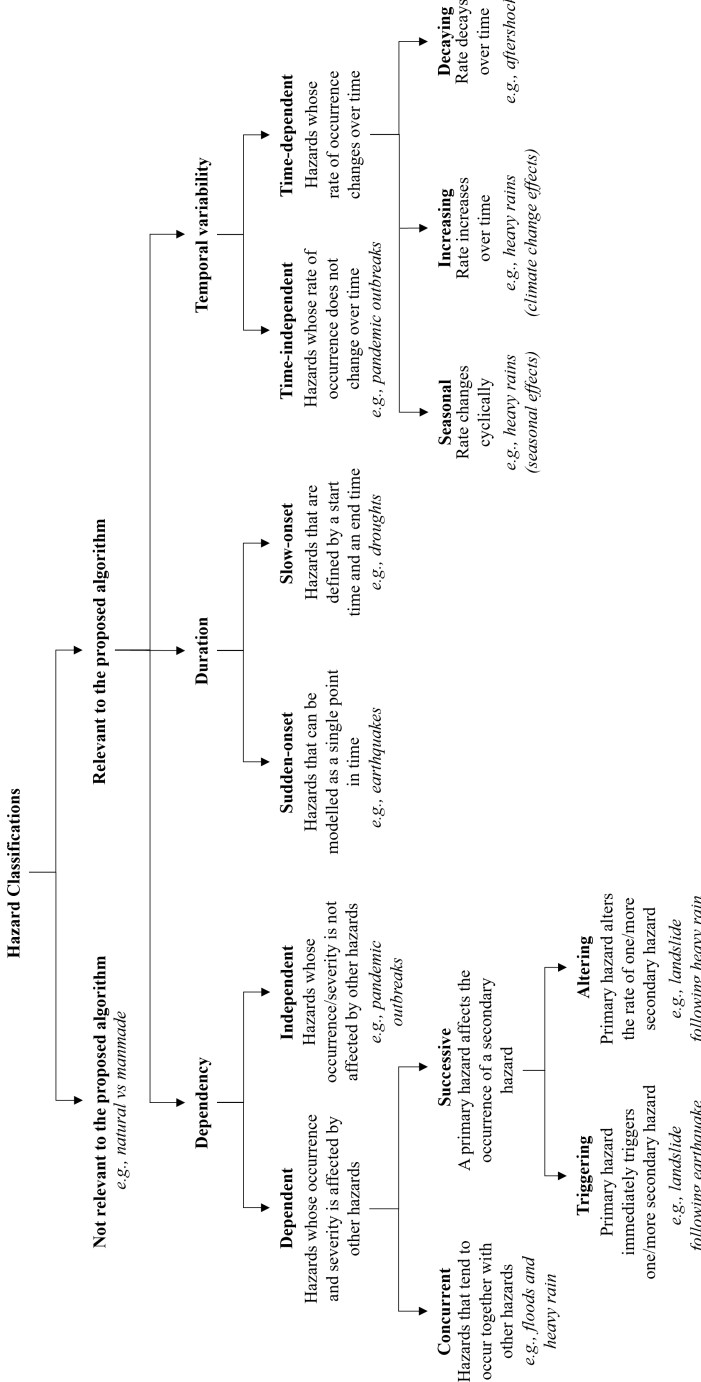

**Figure 1.** Summary of hazard classifications considered in this paper.

## 3.1 Mathematical modeling of event sets

Let us focus first on independent, sudden-onset, time-independent hazards. The discussion will then be extended to incorporate the dependencies across events and to account for slow-onset and time-dependent hazards. The severity measure associated with the occurrence of the $i$-th hazard type $h_i$ is denoted as $m_i$, and the corresponding mean exceedance rate (exceedance rate for brevity hereinafter) is denoted as $\lambda(m_i)$. It must be stressed that, as the hazard types are time-independent, such rates do not change as a function of time. The mean occurrence rate (occurrence rate for brevity hereinafter) of hazard type $h_i$ can be obtained from the rate curve as

$$\lambda_i = \lambda(m_{i,min}) \tag{1}$$

where $m_{i,min}$ is the minimum value of interest of the severity measure (e.g., for earthquakes, it could be the minimum magnitude of engineering interest). In other words, the occurrence rate of $h_i$ is the exceedance rate of its minimum severity measure. A schematic representation of a rate curve can be seen in Figure 2b. If a hazard type is associated with multiple severity measures, rate surfaces define their joint exceedance rate. For example, intensity-duration-frequency surfaces are a standard tool to quantify the mean return period of given rainfall heights and durations (e.g., Fadhel et al., 2017). They define the mean return period (reciprocal to the rate in the case of time-independent hazard types) as a function of both severity measures.

Slow-onset events are defined by the rate curve associated with the start of the event, $\lambda_i^s(m_i)$, and the rate associated with the end of the event, $\lambda_i^e$. The rate curves for time-dependent hazard types are instead described as a function of time.

The rates obtained from these curves/surfaces are used in event simulation, assuming that the event occurrences follow a Poisson process, either homogeneous or non-homogeneous for time-independent and time-dependent hazard types, respectively (for example, the Bartlett-Lewis and the Neyman-Scott models for storm generation in Ritschel et al., 2017). For homogeneous Poisson processes, the interarrival times $t_h$ between event occurrences of hazard type $h_i$ follow an exponential distribution with parameter $\lambda_i$. In this case, simulating a multi-hazard event set consists of randomly sampling numbers from an exponential distribution. A critical assumption of homogeneous processes is that events occur independently of each other, a somewhat restrictive assumption for specific hazard types affected by seasonality and/or previous hazard occurrences (the hazard types classified as dependent and time-dependent in Section 2). To account for such hazard types, we simulate events from non-homogeneous Poisson processes using a procedure known in the literature as thinning (Lewis and Shedler, 1979). This procedure (described in detail in Section 3.3.2) also relies on the random sampling of exponentially distributed numbers and can be efficiently implemented in practice.

## 3.2 Required input

Figure 2a shows a portion of the interaction matrix from Zaghi et al. (2016), which includes flood (F), heavy rain (HR), earthquake (E), and landslide (L). The classification between concurrent and successive interactions is kept unaltered from the original reference. However, the successive interactions have been further separated as triggering (L⟶F, F⟶L, HR⟶L, E⟶L) and altering (E⟶E, L⟶L). The distinction between triggering and altering is needed to capture the different implications of these interactions in the modeling framework.

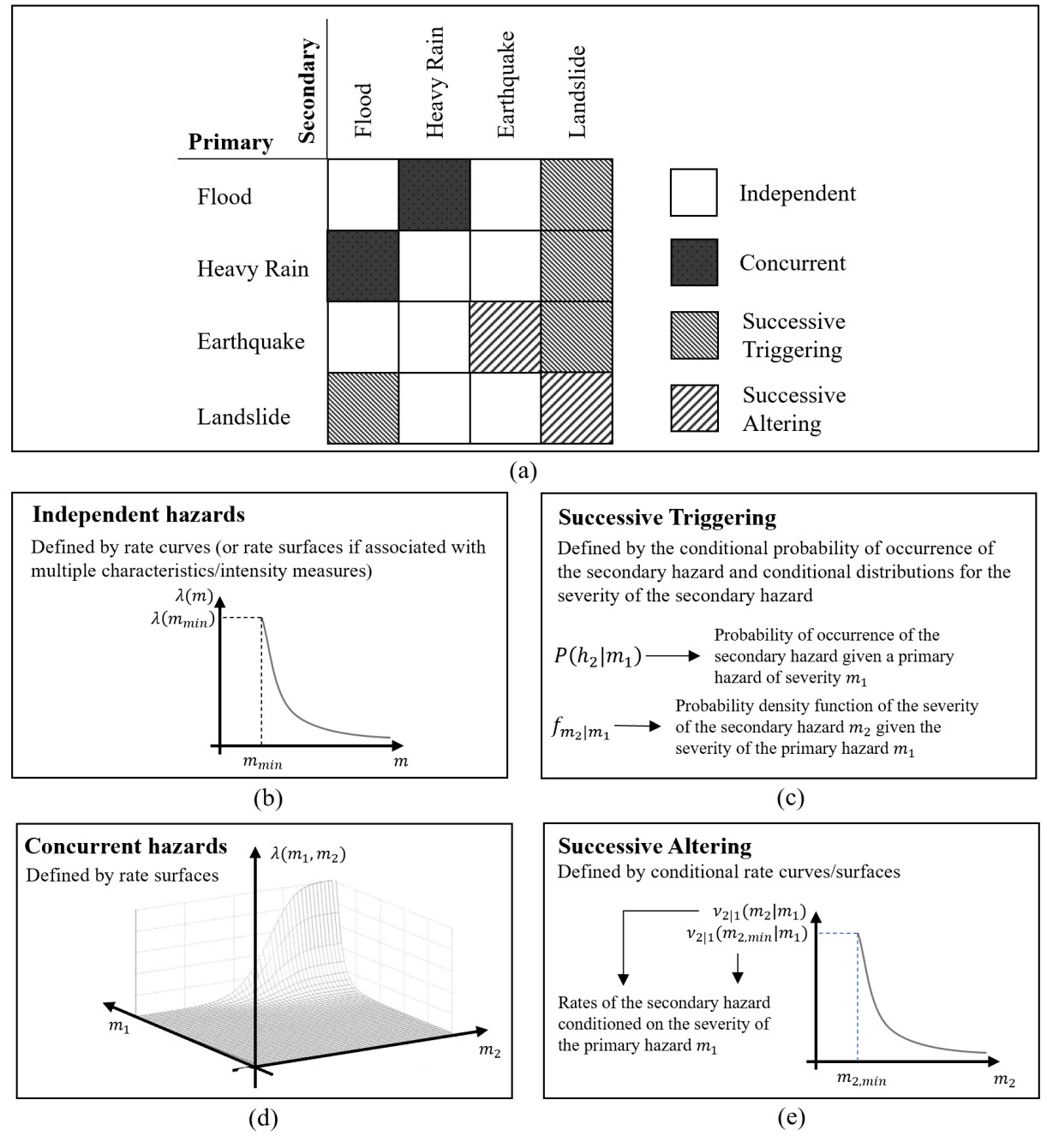

**Figure 2.** Different types of interactions and associated input for the proposed simulation method: (a) hazard classification from Zaghi et al. (2016); (b) rate curve for independent hazard types; (c) conditional probabilities and distributions for successive triggering interactions; (d) rate surfaces for concurrent hazard types; (e) conditional rate curves for successive altering interactions.

Each type of interaction requires different information to be modeled, provided in the list below and shown in Figure 2. For successive interactions, we provide descriptions for the case when $h_2$ is a secondary hazard type following the occurrence of a primary hazard type $h_1$. Depending on the considered interaction, the same hazard type could be classified as primary or secondary (for example, for the set of hazard types in Figure 2, an earthquake could be a mainshock -primary- or an aftershock -secondary-). Also, the discussion can be extended to the case with multiple secondary hazard types.

– *Concurrent interactions* (Figure 2d) are defined by the joint rate of exceedance of the severity measures of all hazard types involved (e.g., the joint exceedance of a given snow depth and a given wind speed, as in Wang and Rosowsky, 2013). This results in rate surfaces that can be interpreted analogously to the ones for single hazard types described by multiple severity measures.

– *Successive - Triggering* (Figure 2c) interactions are defined by the probability of occurrence of $h_2$ conditioned on the severity of $h_1$, i.e., $P(h_2|m_1)$. In cases where the severity measure of $h_2(m_2)$ is of interest, a conditional probability distribution of such quantity ($f(m_2|m_1)$) is also provided (conditioned on the severity measure of $h_1$). As the secondary hazard event(s) is assumed to occur immediately or shortly after the occurrence of the primary hazard event, there is no time component incorporated into the modeling of triggering interactions. An example of conditional probabilities and conditional distributions used to model triggering interactions can be found in Neri et al. (2008), where the probability of several secondary hazard events, such as floods and landslides, is conditioned on the occurrence of the volcanic eruption of Mount Vesuvius. Neri et al. (2008) also provide the variability in the severity measures associated with secondary hazard events. Another example can be found in Parker et al. (2015), where the authors quantify the probability of a landslide on a slope conditioned on the occurrence of an earthquake and its severity.

– *Successive - Altering* (Figure 2e) interactions are defined by the change in the rate curves of $h_2$ following the occurrence of $h_1$. Mainshocks and aftershocks are an example of successive altering interactions. Following a mainshock, the rate of aftershocks is typically modified in terms of the characteristics of the mainshock using the modified Omori law (Utsu, 1970) or more advanced models (e.g., Iacoletti et al., 2022). Rates for this type of interaction typically decay with time as the memory of the primary hazard subsides, resulting in non-homogenous Poisson processes. This paper proposes a thinning methodology to incorporate non-homogeneous processes into the formulation (Section 3.3.2). Alternatively, the non-homogeneous processes can be translated into an equivalent homogeneous process with a given memory $mem_i$. By taking the inverse of such memory, we can also define a "memory loss rate", $\zeta_i = 1/mem_i$. This rate determines how often the system loses its memory of $h_1$ and the occurrence rates of $h_2$ return to the original level (e.g., zero for aftershock occurrences). We call rate curves unaffected by altering interactions *original rate curves* and rate curves affected by altering interactions *conditional rate curves*.

## 3.3 Life cycle hazard event set simulation

We incorporate the above information into a sequential MC simulation approach. The procedure outputs life cycle hazard event sets, i.e., multi-hazard event sets (times of occurrence, hazard types, and associated severity measures) throughout the

life cycle of the system of interest. Because of the simplifying assumptions, the simulation of such event sets is computationally efficient while retaining the relevant implications of the hazard interactions. It can be repeated multiple times to obtain relevant statistical quantities such as (i) the probability of having a given number of occurrences of a specific hazard type in a given time span; (ii) the probability of occurrence of a specific combination of hazard events within the life cycle; and (iii) the probability distribution of the severity measures of the hazard events and joint probability distributions of the severity measures of multiple hazard events occurring within a short time frame. These quantities, as well as the simulated event sets, can be incorporated into formulations for Level II interactions (e.g., Selva, 2013; Dehghani et al., 2021; Otárola et al., 2023a, b) to obtain the expected consequences of the hazard events throughout the life cycle of a system. To the best of the authors' knowledge, no algorithm is currently available in the literature that accounts for the types of interactions and additional aspects (i.e., event duration and temporal variability) highlighted in this paper. An alternative sequential MC approach has been proposed by Mignan et al. (2014), which separates the simulation of primary events from the simulation of secondary events (all primary events are simulated, then all secondary events are simulated). However, such an algorithm only considers sudden-onset, time-independent hazard events, and models all interactions as successive, triggering. Similarly, Selva (2013) used simplified, closed-form solutions to translate rate curves into probabilities of occurrence of the hazards within a given time period. While the interactions between hazards can be included by modifying the probability of occurrence of the secondary hazard (Selva introduces "co-active risk factors" for this purpose, and provides an example with volcanic eruptions and ash fallout), such an approach is also limited to sudden-onset, time-independent events and does not capture the intricacies of hazard sequences that may include multiple successive interactions.

The following subsections describe the proposed algorithm to generate the event sets, starting with the case with time-independent, sudden-onset hazard types, and then extending the discussion to time-dependent and slow-onset hazard types. It is worth noting that, in the presence of a single hazard (e.g., earthquakes) from different sources, the occurrence of the event from each source can be modeled as its own hazard type (e.g., $h_1$ is earthquake on source 1, $h_2$ is earthquake from source 2...) each characterized by a given rate of occurrence (and characteristics).

### 3.3.1 Proposed algorithm

Figure 3 shows a visual representation of the proposed algorithm to simulate event sets for the case with time-independent, sudden-onset hazard types. The flowchart in Figure 4 details the described sequential MC approach, with reference to each step shown in Figure 3. Every type of interaction is included in the proposed procedure and incorporated based on its specific characteristics.

The simulation of the event set starts in a neutral state where the rates for each hazard type $h_i (i = 1, \ldots, N)$ are defined based on the corresponding original rate curves (Step 1 in Figure 3). We define $\boldsymbol{I}$ as the set of indices of the hazard types that, at any given time, have affected the rate of any secondary hazard type (e.g., if the system retains memory of hazards $h_1$ and $h_3$, we will have $\boldsymbol{I} \equiv \{1, 3\}$). Because the system has no memory of any previous hazard events at this point, we set $\boldsymbol{I} = \varnothing$. The theory of competing Poisson processes determines that the rate of occurrence of the first hazard event is equal to the sum of the rates of the individual hazard types. Consequently, the time of occurrence $t$ of the first hazard event is sampled from an exponential

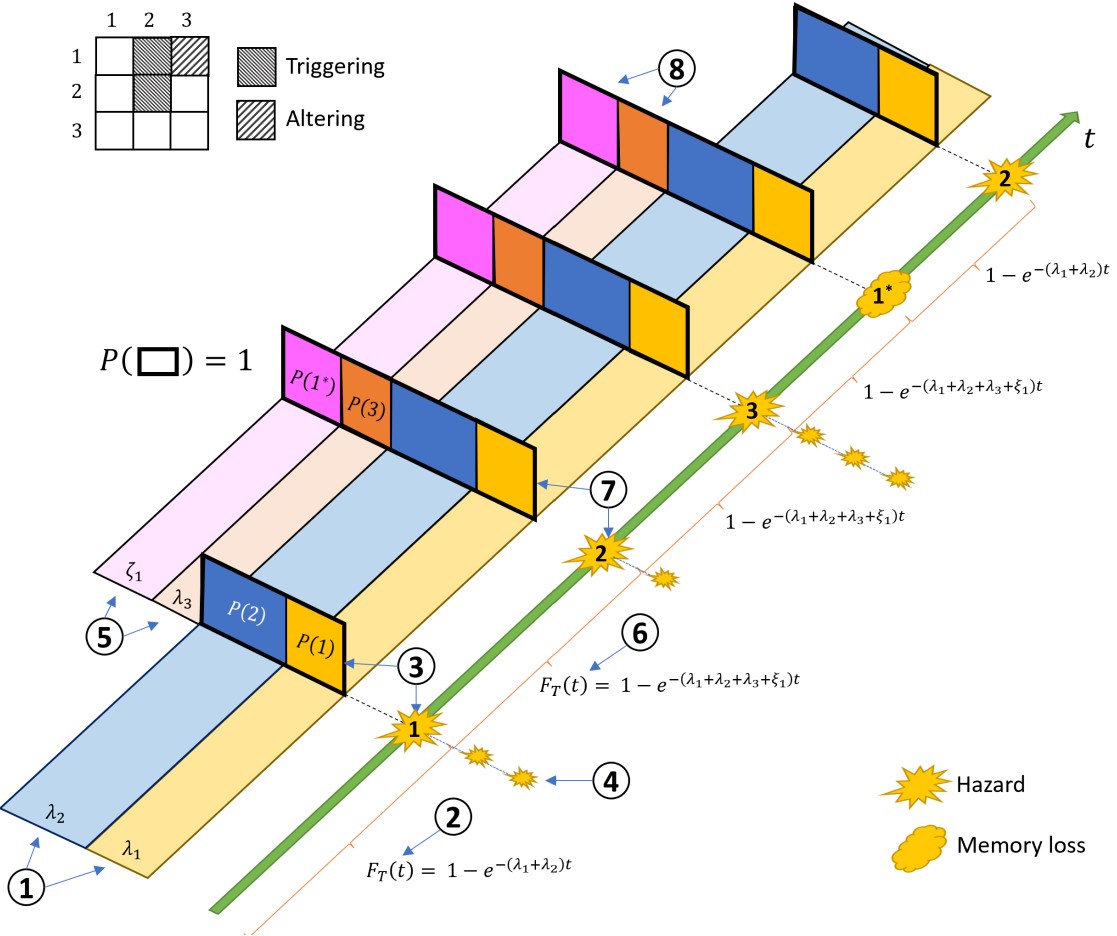

**Figure 3.** Visual representation of a simulated life cycle hazard event set. Step 1: the rates of all hazard types are established (the original rate of hazard type 3 is $= 0$ in this figure); Step 2: an event is simulated; Step 3: The hazard type 1 is assigned to the event; Step 4: Hazards due to successive triggering interactions are simulated; Step 5: Based on the identified successive altering interactions, the rate of hazard type 3 is modified and the memory loss event for hazard type 1 is introduced; Step 6: A new event is simulated; Step 7: The hazard type 2 is assigned to the event; Step 8: After a memory loss event is simulated, the rate of hazard type 3 is set to the original value $(= 0)$ and the memory loss of hazard type 1 is removed from the pool of possible events.

distribution ($f_T(t) = \lambda e^{-\lambda t}, t > 0$) with parameter $\lambda = \sum_{n=1}^{N} \lambda_n$ (Step 2 in Figure 3). Once the event has been simulated, it is assigned to one of the $i$-th hazard types $h_i$ (Step 3 in Figure 3). The probability that the hazard event corresponds to the $i$-th hazard type $h_i$ is

$$P(H = h_i) = \frac{\lambda_i}{\sum_{n=1}^{N} \lambda_n} \tag{2}$$

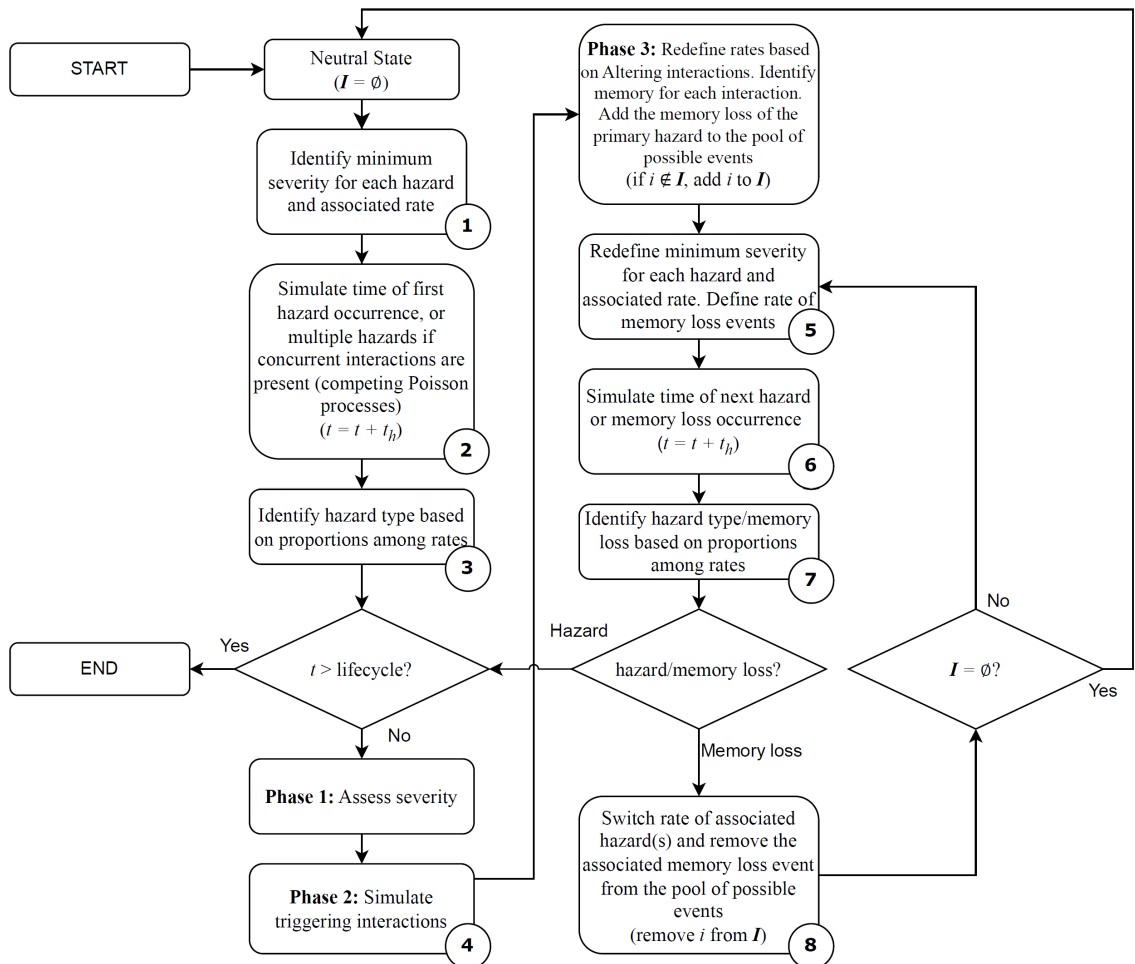

**Figure 4.** Flowchart of the proposed simulation method.

Three phases follow the simulation of a hazard event: Phase 1 is the assessment of the hazard severity (i.e., $m_i$); Phase 2 is the simulation of hazard events caused by triggering interactions; and Phase 3 is the reassessment of the rates based on altering interactions.

In Phase 1, the severity measure of the simulated hazard event $m_i$ is obtained from the rate curve of the $i$-th hazard type $\lambda_i(m_i)$ using the one-to-one relationship between the Cumulative Distribution Function (CDF) of the random variable $M_i$ and its rate of exceedance, as

$$F_{M_i}(m_i) = 1 - \frac{\lambda_i(m_i)}{\lambda_i(m_{min})} \tag{3}$$

In the case of hazard types associated with multiple severity measures and/or concurrent hazards, all severity measures are
obtained from rate surfaces rather than rate curves. A similar relationship to Eq. (3) can be obtained for the multi-dimensional case. Phase 1 is not shown in Figure 3 for visual clarity.

In Phase 2, the triggering interactions are simulated (Step 4 in Figure 3). The occurrence of each secondary hazard event is simulated by considering the conditional probability established within the triggering interaction's definition. Subsequently, the severity measure of the secondary event(s) is sampled from the corresponding conditional probability distribution.

In Phase 3, the rates of each hazard type are reassessed to account for the altering interactions. In particular, for each altering interaction (Step 5 in Figure 3): (i) we substitute the original rate curves for each secondary hazard type with the corresponding conditional rate curve; (ii) we introduce an additional "memory loss" Poisson event with rate $\zeta_i = 1/mem_i$ (associated with the primary hazard event) to the pool of possible events; (iii) if $i \notin I$, we add $i$ to the set of indices $I$.

We can then simulate the following event, which can be either the occurrence of a new hazard event (with rate $\lambda_i$) or a memory loss event (with rate $\zeta_i$). The theory of competing Poisson processes determines that the rate of the next event occurrence is equal to the sum of the rates of the individual events (which now include both hazard events and memory loss events). Consequently, the time of occurrence of the next event is sampled from an exponential distribution with parameter $\lambda = \sum_{n=1}^{N} [\lambda_n + \zeta_n \mathbf{1}_{\{n \in I\}}]$ (Step 6 in Figure 3), where $\mathbf{1}_{\{\bullet\}}$ is the indicator function (i.e., $\mathbf{1}_{\{n \in I\}} = 1$ if $n \in I$ and $\mathbf{1}_{\{n \in I\}} = 0$ if $n \notin I$). The simulated event is then assigned to either one of the hazard types ($\{H = h_i\}$) or with the loss of memory of one of the hazard types ($\{H = h_i^*\}, i \in I$) (Step 7 in Figure 3). The probability that the next event is the occurrence of the $i$-th hazard type ($\{H = h_i\}$) is

$$P(H = h_i) = \frac{\lambda_i}{\sum\limits_{n=1}^{N} [\lambda_n + \zeta_n \mathbf{1}_{\{n \in I\}}]} \tag{4}$$

and the probability that the next event to occur is a loss of memory of the $i$-th hazard type ($\{H = h_i^*\}, i \in I$) is

$$P(H = h_i^*) = \frac{\zeta_i}{\sum\limits_{n=1}^{N} [\lambda_n + \zeta_n \mathbf{1}_{\{n \in I\}}]} \tag{5}$$

If the simulated event is a hazard event, Phases 1-3 are repeated. If the simulated event is the memory loss of the $i$-th hazard type, we remove $i$ from the set of indices $I$, replace the corresponding conditional rate curves with the original rate curves, and remove the Poisson event with rate $\zeta_i$ from the pool of possible events (Step 8 in Figure 3).

### 3.3.2 Incorporating time-dependent events

This section details how to modify the procedure in Section 3.3.1 to incorporate non-homogeneous Poisson events used to capture the temporal variability of the hazard (time-dependent events in Section 2). We focus on the case with multiple decaying processes associated with altering interactions (e.g., aftershocks after the occurrence of mainshocks). The procedure can be adapted to the case of seasonal processes with slight modifications. Figure 5 visualizes the described procedure. To aid in the interpretation of Figure 5, the reader may think of hazard type 1 as mainshocks and hazard type 3 as aftershocks.

In this case, we define $J$ as the set of indices of the (secondary) hazard types that have been affected by altering interactions. No memory loss events need to be considered in this case. Because all the considered non-homogeneous processes follow from altering interactions, and the original rate curves are associated with homogeneous processes, Steps 1-4 described in Section

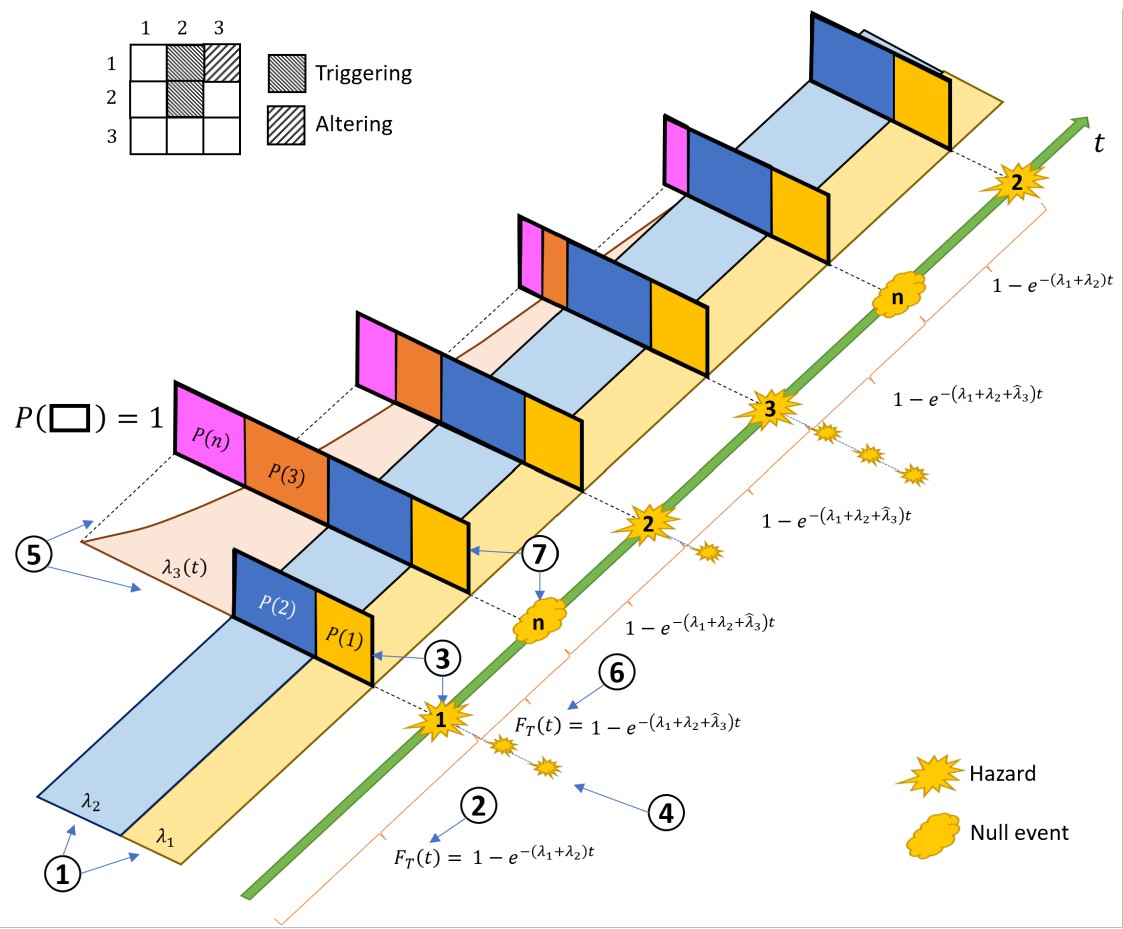

**Figure 5.** Visual representation of the proposed simulation method in the presence of time-dependent (non-homogeneous) processes. Step 1: the rates of all hazard types are established (the original rate of hazard type 3 is $= 0$ in this figure); Step 2: an event is simulated; Step 3: The hazard type 1 is assigned to the event; Step 4: Hazard events due to successive triggering interactions are simulated; Step 5: Based on the identified successive altering interactions, the rate of hazard type 3 is modified. For the purpose of event simulation, the rate of hazard type 3 is fixed to its maximum value; Step 6: A new event is simulated; Step 7: The simulated event is discarded (i.e.,it is classified as a null event).

3.3.1 are unchanged. After the occurrence of a primary hazard event associated with at least one altering interaction, we define the time-varying rate of the non-homogeneous process associated with the $i$-th (secondary) hazard type as $\lambda_i(t - t_0)$ (for each of the secondary hazards of the altering interactions), where $t_0$ is the time of occurrence of the primary hazard event that caused the altering interaction (Step 5 in Figure 5). We also add the indices associated with the secondary hazard types to $\boldsymbol{J}$, if they were not already included in the set (i.e., $i \notin \boldsymbol{J}$). To simulate the occurrence of the next hazard event, which results from multiple competing homogeneous and non-homogeneous processes, we use a modified, sequential version of a procedure known in the literature as thinning, which involves using a higher, homogeneous rate for event simulation, and then discarding a selection of the simulated events, i.e., classifying them as null events (Lewis and Shedler, 1979). In the proposed sequential

approach, for each $i$ in $\boldsymbol{J}$, we fix the rate of the associated non-homogeneous process at its maximum value, i.e., $\hat{\lambda}_i = \lambda_i(0)$. We then generate the interarrival time to the next event $t_h$ ($t_h = t^* - t_0$, where $t^*$ is the time of occurrence of the event) from an exponential distribution with parameter $\lambda = \sum_{n=1}^{N}[\lambda_n \mathbf{1}_{\{n \notin \boldsymbol{J}\}} + \hat{\lambda}_n \mathbf{1}_{\{n \in \boldsymbol{J}\}}]$ (Step 6 in Figure 5). The simulated event is then assigned to either one of the $N$ hazard types ($\{H = h_i\}$) or to a null event from the non-homogeneous Poisson process associated with the $i$-th hazard type ($\{H = n_i\}, i \in \boldsymbol{J}$) (Step 7 in Figure 5). The probability that the next event is the occurrence

of the $i$-th hazard type ($\{H = h_i\}$) is

$$P(H = h_i) = \begin{cases} \dfrac{\lambda_i}{\sum\limits_{n=1}^{N} [\lambda_n \mathbf{1}_{\{n \notin \boldsymbol{J}\}} + \hat{\lambda}_n \mathbf{1}_{\{n \in \boldsymbol{J}\}}]} & \text{if } i \notin \boldsymbol{J} \\[4mm] \dfrac{\lambda_i(t^* - t_0)}{\sum\limits_{n=1}^{N} [\lambda_n \mathbf{1}_{\{n \notin \boldsymbol{J}\}} + \hat{\lambda}_n \mathbf{1}_{\{n \in \boldsymbol{J}\}}]} & \text{if } i \in \boldsymbol{J} \end{cases} \tag{6}$$

and the probability that the next event to occur is a null event from the non-homogeneous Poisson process associated with the $i$-th hazard type ($\{H = n_i\}, i \in \boldsymbol{J}$) is

$$P(H = h_i) = \frac{\hat{\lambda} - \lambda_i(t^* - t_0)}{\sum\limits_{n=1}^{N} [\lambda_n \mathbf{1}_{\{n \notin \boldsymbol{J}\}} + \hat{\lambda}_n \mathbf{1}_{\{n \in \boldsymbol{J}\}}]} \tag{7}$$

If the simulated event is a hazard event, Phases 1-3 described in Section 3.3.1 are performed (with the possible introduction of additional, non-homogeneous Poisson processes). If the simulated event is a null event, it is discarded from the hazard event set. In both cases, we set $t_0 = t^*$, and we update the rate of the non-homogeneous Poisson process(es) to $\hat{\lambda}_i = \lambda_i(t_0 - \hat{t}_0)$, where $\hat{t}_0$ is the time of occurrence of the event that started the non-homogeneous Poisson process. Given the decaying nature of the rates of the non-homogeneous processes considered herein, we assume that the effects of the altering interactions are

forgotten whenever the difference between the updated rate of the process $\hat{\lambda}_i$ and the rate of the process $\lambda_i$ from the original rate curve falls below a pre-specified threshold $\epsilon$ (i.e., if $\hat{\lambda}_i - \lambda_i < \epsilon$). The value of $\epsilon$ depends on the specific hazard, and it is subjective. Generally, $\epsilon$ can be selected based on experience or engineering judgment. Whenever $\hat{\lambda}_i - \lambda_i < \epsilon$ in the sequential MC procedure, we remove $i$ from $\boldsymbol{J}$, and we revert to using the original rate curve. A flowchart in Appendix A details the described sequential MC approach, with reference to each step shown in Figure 5.

### 330  3.3.3  Incorporating slow-onset events

This section details how to modify the procedure in Section 3.3.1 to incorporate possible slow-onset events. Figure 6 visualizes the described procedure. The simulation of the $i$-th slow-onset event is herein modeled with a two-step approach. First, the event's start is simulated using a homogeneous Poisson process with a distinct start rate $\lambda_i^s$. Then, the start rate is replaced by an end rate $\lambda_i^e$, which is used to model the conclusion of the event (with a corresponding homogeneous Poisson process). The rate

change could be interpreted as an altering interaction of the hazard with itself. This method facilitates a realistic representation of events characterized by varying duration. The severity of such events is assumed to be non-varying throughout the event. However, the procedure could be modified to include events with time-varying severity. Slow-onset events whose presence could also affect the rate of additional processes (e.g., a drought affecting the rates of wildfires and floods) can be modeled by

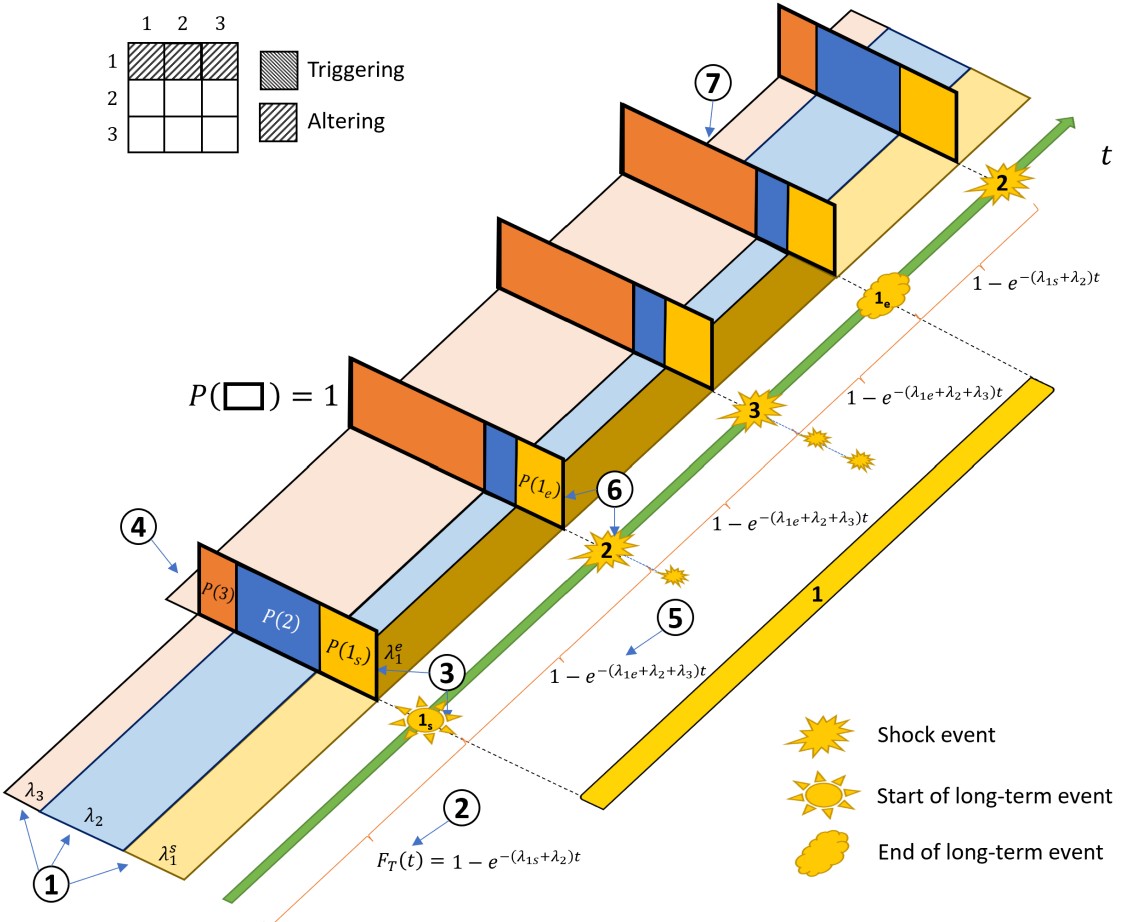

**Figure 6.** Visual representation of the proposed simulation method in the presence of slow-onset events. Step 1: the rates of all hazard types are established. For slow-onset events (hazard type 1), the start rate is selected; Step 2: an event is simulated; Step 3: The event is assigned to the start of hazard type 1; Step 4: Rates are modified based on successive altering interactions; Step 5: A new event is simulated; Step 6: The hazard type 2 is assigned to the event; Step 7: After simulating the end of a slow-onset event, the rates of the secondary hazard types are set to the original values.

assigning a successive altering interactions to the start of the event (e.g., the start of the slow-onset event causes a change in the rates associated with the secondary hazard types) and by associating the end of the event with the loss of memory of the interactions (i.e., the end of the slow-onset event effectively acts as a memory loss event, see Section 3.3.1). As the memory loss is marked by the end of the slow-onset event, no additional memory-loss event needs to be added in this case. To aid in the interpretation of Figure 6, the reader may interpret hazard type 1 as droughts (slow-onset), hazard type 2 as floods (rate decreases during droughts), and hazard type 3 as wildfires (rate increases during droughts).

A flowchart in Appendix A details the described sequential MC approach, with reference to each step shown in Figure 6.

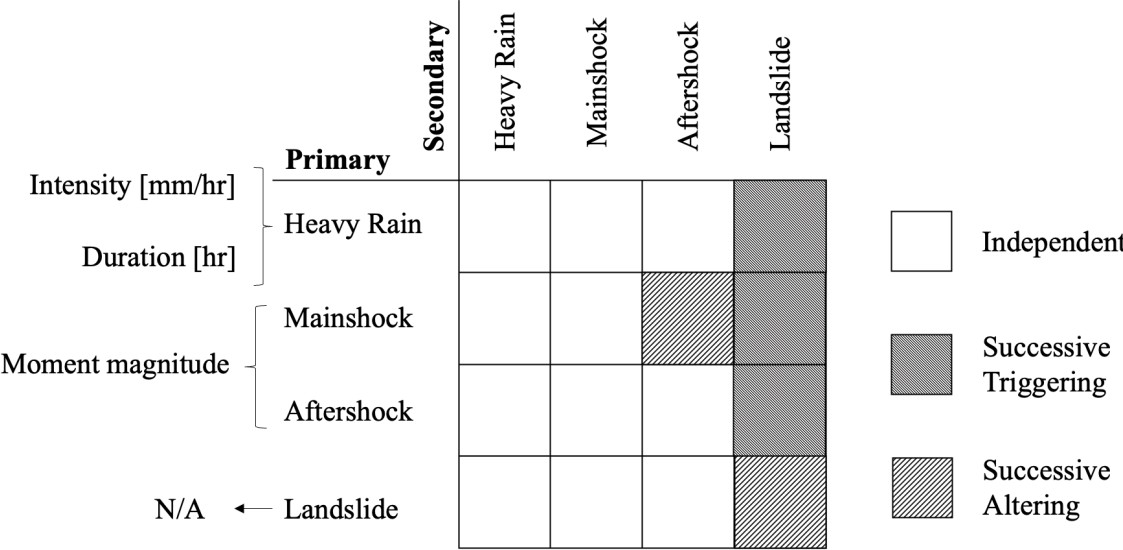

**Figure 7.** Taxonomy of interactions for the case study and associated input parameters.

## 4 Numerical Example

We now showcase the life cycle hazard event sets simulation using the sequential MC method detailed in the previous sections. Figure 7 shows the hazard types considered in the example and their interaction. Figure 7 also reports the severity measure(s) adopted for the considered hazard types. There are no concurrent interactions considered in this example. However, the hazard type "Heavy Rain" is associated with multiple severity measures (i.e., intensity and duration), and the modeling of single hazards with multiple associated severity measures is equivalent to the modeling of concurrent hazards (see Section 3).

Earthquakes have been separated into mainshocks and aftershocks for modeling purposes. This distinction allows us to separate the rate curves for the two hazard types, with aftershocks having a rate $= 0$ before the occurrence of a mainshock and a conditional rate curve after the occurrence of a mainshock. The distinction also allows the system to retain a memory of the mainshock after the occurrence of the aftershocks. Without this distinction, the simulated occurrence of the first aftershock would redefine the rates of subsequent aftershocks, and the effects of the mainshock would be forgotten. More sophisticated models where the occurrence of early aftershocks in the sequence affects the rate of subsequent aftershocks can also be found in the literature (Ogata, 1998) but are not selected here for illustrative purposes. In the following, we report the reference for each of the adopted rate/surface curves for single hazards and each of the interaction models (in this order). Such references are also summarized in Table 1.

We note that the curves/surfaces/models used in this numerical example have been developed for different locations worldwide. As such, the event sets obtained in this example shall not be associated with any specific location. The example showcases the potential of the proposed simulation method and shows that the required information can be retrieved from the literature.

**Table 1.** Hazard curves/surfaces and interaction models for the numerical example.

| Input | Reference | Additional details |
|---|---|---|
| Rate curves for earthquakes | Iervolino et al. (2018) | Zone 923 |
| Rate surfaces for heavy rain | Tang and Cheung (2011) | Rain gauge N05 |
| Mainshock/aftershock | Yeo and Cornell (2009) | Parameters from Iervolino et al. (2018) |
| Earthquake/landslide | Parker et al. (2015) | Model for the 1968 earthquake |
| Heavy rain/landslide | Liu and Wang (2022) | Non-stabilized slope model |
| Landslide/landslide | Samia et al. (2017) | - |

The collection of information for a specific location (which would require a tailored investigation/literature review) is outside

the scope of the paper.

**Mainshocks/aftershocks:** The rate curve for the occurrence of the mainshock events, $\lambda_m(m_m)$, is obtained from Iervolino et al. (2018). Namely, we use the exceedance curves for Zone 923 of the Italian earthquake hazard model, which corresponds to the L'Aquila region. The severity measure of earthquakes is expressed in terms of moment magnitude $M_w$ ($M_m$ for the mainshocks and $M_a$ for the aftershocks). The minimum severity measure for both mainshocks and aftershocks is assumed as

$m_{m,min} = m_{a,min} = 4.45$, which is slightly higher than the one in the original reference (4.15). This is to reduce the number of small earthquakes in the simulated life cycle hazard event set to improve the clarity of this illustrative application. Aftershocks cannot occur independently from a mainshock. Therefore, their original rate curve is set to $\lambda_a(m_a) = 0 \forall m_a$. All earthquakes are assumed to occur at the same location (or very close in space), such that the shortest Joyner-Boore distance (Joyner and Boore, 1981) of each earthquake from the location of interest is $R_{jb} = 20$ km.

**Heavy rain:** The rate surface for the occurrence of heavy rain events, $\lambda_r(m_{r,1}, m_{r,2})$, is obtained from the Intensity-Duration Frequency curves reported by Tang and Cheung (2011) for the Hong Kong region. Namely, we use the data for the rain gauge N05 at Cheung Chi House. The severity measures associated with the heavy rain events are the duration ($M_{r,1}$) expressed in hours (hr) and the intensity ($M_{r,2}$) expressed in millimeters per hour (mm/hr). The minimum considered severity measures are $m_{r,1,min} = 0.083$ hr (5 minutes) and $m_{r,2,min} = 0.893$ mm/hr for the duration and the intensity, respectively.

**Landslide:** We assume that landslide events cannot occur independently for this case study. They can only occur as secondary events of a successive interaction with earthquakes, heavy rain, and/or previous landslides. We also assume for simplicity that landslide events are not associated with any severity measure. The details of the above-mentioned interactions are detailed below.

**Mainshock/aftershock interactions:** The interaction between mainshocks and aftershocks is modeled as a successive, al-

385 tering interaction following the modified Omori law (Yeo and Cornell, 2009). According to the Omori law, after the occurrence of a mainshock of magnitude $m_m$ at time $t_m$, aftershocks occur following a non-homogeneous Poisson process with a time-varying rate

$$\lambda_a(t) = \frac{10^{a+b(m_m - m_{m,min})} - 10^a}{[(t - t_m) + c]^p} \tag{8}$$

where $t$ and $t_m$ quantities are expressed in days, and $a,b,c$, and $p$ are the parameters of the model. For this example, we assume (from Iervolino et al., 2018) $a = -1.66$, $b = 0.96$, $c = 0.03$, and $p = 0.93$. The severity of the aftershocks is simulated from the rate curve used for the mainshock (using Eq. 3).

**Earthquake(mainshock/aftershock)/landslide interactions:** The interaction among earthquakes and landslides is modeled as a successive, triggering interaction following the model in Parker et al. (2015), calibrated based on data from the South Island region in New Zealand. The probability of a landslide given the occurrence of an earthquake with severity measure $m_w$, $P_0(L|m_w)$, is

$$P_0(L|m_w) = \frac{1}{1 - e^{-(c_0 + c_{SL}SL + c_{NDS}NDS + c_{PGA}PGA(m_w, R_{jb}))}} \tag{9}$$

where $SL$ is the local hillslope gradient (we assume $SL = 35°$ for the slope of this case study), $NDS$ is the normalized distance from stream to ridge crest (we assume $NDS = 0.5$ for this case study), $c_0, c_{PGA}, c_{SL}$, and $c_{NDS}$ are parameters of the logistic regression used to fit Eq. 9 to the data, and $PGA(m_w, R_{jb})$ is the peak ground acceleration caused by the earthquake at the location of interest. We obtain $PGA(m_w, R_{jb})$ using the ground motion model in Huang and Galasso (2019), assuming that the soil type is rock and that the style-of-faulting of the earthquake is strike-slip fault:

$$PGA(m_w, R_{jb}) = b_1 + b_2 m_w + b_3 m_w^2 + (b_4 + b_5 m_w)\log_{10}\left(\sqrt{R_{jb}^2 + b_6^2}\right) \tag{10}$$

where $\mathbf{b} = [b_1, b_2, b_3, b_4, b_5, b_6]$ is a vector of unknown parameters. From the PGA model with spatial correlation in Huang and Galasso (2019), we select $b_1 = 3.524$, $b_2 = 0.247$, $b_3 = -0.020$, $b_4 = -3.936$, $b_5 = 0.351$, and $b_6 = 12.417$.

We note that the $PGA(m_w, R_{jb})$ obtained from Eq. 10 is the median value at the location of interest, and a more refined analysis should consider the uncertainties associated with this value. However, a full probabilistic analysis of the ground motion is outside the scope of this example.

**Heavy rain/landslide interactions:** The interaction among heavy rain events and landslides is modeled as a successive, triggering interaction following the model in Liu and Wang (2022). Namely, the probability of a landslide after a rainfall of intensity $m_{r,2}$ is given by

$$P_0(L|m_{r,2}) = \begin{cases} 1 & \text{if } m_{r,1} > D_c(m_{r,2}) \\ 0 & \text{if } m_{r,1} < D_c(m_{r,2}) \end{cases} \tag{11}$$

where $D_c(m_{r,2})$ is the critical rainfall duration for slope instability associated with the intensity $m_{r,2}$. We use the critical rainfall duration for the slope before stabilization in Liu and Wang (2022), i.e.

$$D_c(m_{r,2}) = 2.61 \times 10^4 m_{r,2}^{-2.030} + 27.04 \tag{12}$$

**Landslide/landslide interactions:** The interactions among subsequent landslides are modeled as a successive, altering interaction following the model in Samia et al. (2017). The paper reports that the susceptibility of a slope to a landslide ($s_l$) increases by 15-fold immediately after the occurrence of a previous landslide, and then it decreases exponentially over time

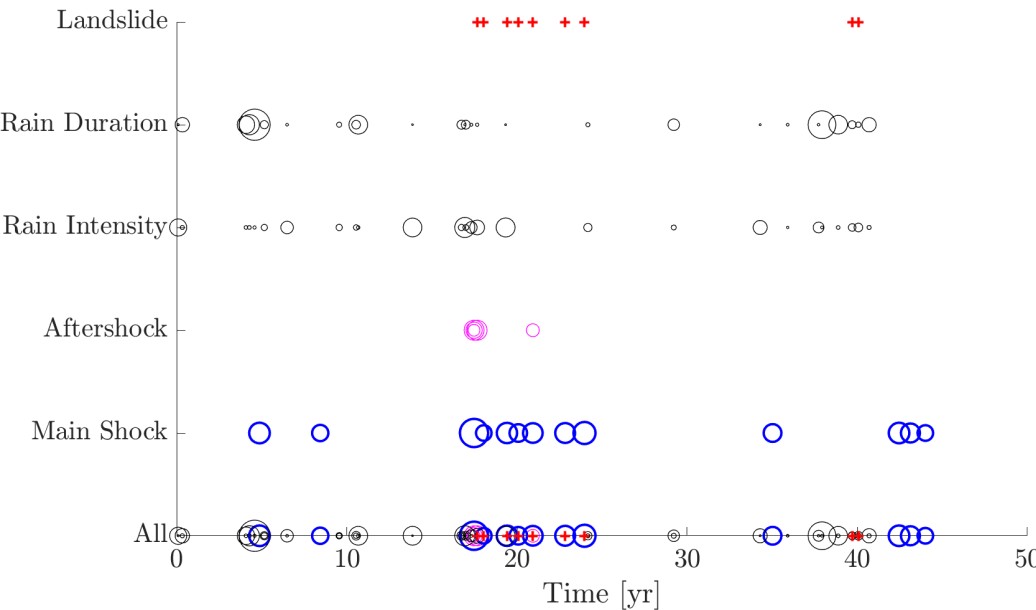

**Figure 8.** Example simulated life cycle hazard event set. Note: the radius of the circles is indicative of the event's severity.

with an exponential coefficient value of $c_l = -0.12$. Hence, we define the time-varying susceptibility of the slope as

$$s_l(t) = \begin{cases} 1 & \text{if } t < t_{l1} \\ 15e^{-0.12(t-t_{ll})} & \text{if } t > t_{l1} \end{cases} \tag{13}$$

where $t_{l1}$ is the time of occurrence of the first landslide event and $t_{ll}$ is the time of occurrence of the latest landslide event. Consequently, in the aftermath of a landslide event, we modify the probability of occurrence of landslides due to earthquakes and heavy rain events as $P(L|m_w) = s_l(t)P_0(L|m_w)$ and $P(L|m_{r,1}, m_{r,2}) = s_l(t)P_0(L|m_{r,1}, m_{r,2})$, respectively.

Figure 8 shows an example event set generated using the proposed method. The severity measure of the events is proportional to the diameter of the circles used to represent their occurrence. As expected, more severe mainshocks produce more severe

aftershock sequences. The rate and severity measures (intensity and duration) of heavy rain events are generated based on their joint rate surface. Sequences of landslide events can also be found around years 17-25 in the system's life cycle, triggered mostly by mainshock-aftershock sequences, and at year 40, triggered by heavy rain events.

Simulating a multi-hazard event set such as the one shown in Figure 8 is computationally efficient since it amounts to sampling numbers from exponential distributions. Consequently, multiple realizations can be used to obtain relevant statistics

for the system's life cycle. For example, Table 2 reports the mean and median number of events, categorized by hazard type, during the system's life cycle, obtained from 25,000 simulations. The difference between the mean and median value can serve as an effective indicator of the skewness of the associated distribution of events. When the mean and median are close, the distribution exhibits a scarcity of outliers, with realizations evenly distributed around the mean. Conversely, a notable difference

between the mean and median signals heightened variability among realizations, suggesting the presence of multiple event sets with either a scarcity (e.g., zero) or a high number of events.

**Table 2.** Hazard curves/surfaces and interaction models for the numerical example.

| Hazard type | Mean number of events | Median number of events |
|---|---|---|
| Heavy Rain | 25.86 | 26 |
| Mainshock | 12.02 | 12 |
| Aftershock | 36.02 | 17.5 |
| Landslide | 17.36 | 8 |

The simulated event sets may be used to identify the possible occurrence of independent hazard in close temporal proximity. To that purpose, Figure 9 shows the mean (Figure 9a) and median (Figure 9b) number of pairwise hazard combinations throughout the life cycle of the system. A hazard combination is identified whenever two hazards occur with an interarrival time < 200 days, and no other hazard event occurs between them. The threshold of 200 days is arbitrary, and it has been selected solely for demonstration purposes. Nevertheless, it effectively represents a reasonable temporal interval that would pose challenges to the recovery from the initial hazard event before the occurrence of the subsequent hazard event (e.g., Opabola and Galasso, 2024). Heat maps such as the one in Figure 9 can help identify hazard combinations that may pose significant challenges in terms of possible consequences on the physical assets of interest (which would require the modeling of Level II interactions). It can be observed that even hazard types that are not related by level I interactions may occur close (in time) to each other. For example, there are, on average 2.62 main shock events following the occurrence of a heavy rain event and 1.47 heavy rain events following a mainshock event, which might suggest that the interactions between such (independent) hazards might be of interest in a possible life cycle analysis of a structure placed in the investigated location. This sort of 'coincidental' hazard combinations should not surprise the end users of the algorithm. In fact, such combinations have been observed on multiple occasions over the past century and are expected to increase due to climate change (e.g., Cutter et al., 2008). For example, typhoons were recorded in close temporal proximity to the great Kanto earthquake (Japan) of 1923 (e.g., Sasaki and Yamakawa, 2007) and the Hokkaido earthquake (Japan) of 2018 (e.g., Heidarzadeh et al., 2023). Although less frequent, these combinations are just as crucial as those influenced by causality (de Ruiter et al., 2020). An additional advantage of the proposed method is that it seamlessly integrates causal and coincidental event combinations within the same formulation.

Finally, joint distributions for the severity measures of multiple hazards can be obtained for each hazard combination of interest. For example, Figure 10 shows the joint probability density function (obtained using the kernel density method) of the severity measures for the hazard combinations mainshock-mainshocks (Figure 10a), mainshock-aftershock (Figure 10b), and mainshock-heavy rain (Figure 10c). As expected, hazard combinations that are linked by Level I interactions show clear signs of correlation (mainshock-aftershock). In contrast, the joint PDF of the severity measures for independent hazards shows no apparent signs of correlation and can be approximated by the product of the marginal PDFs of the severity measures for the individual hazards (which can be obtained from Eq. 3).

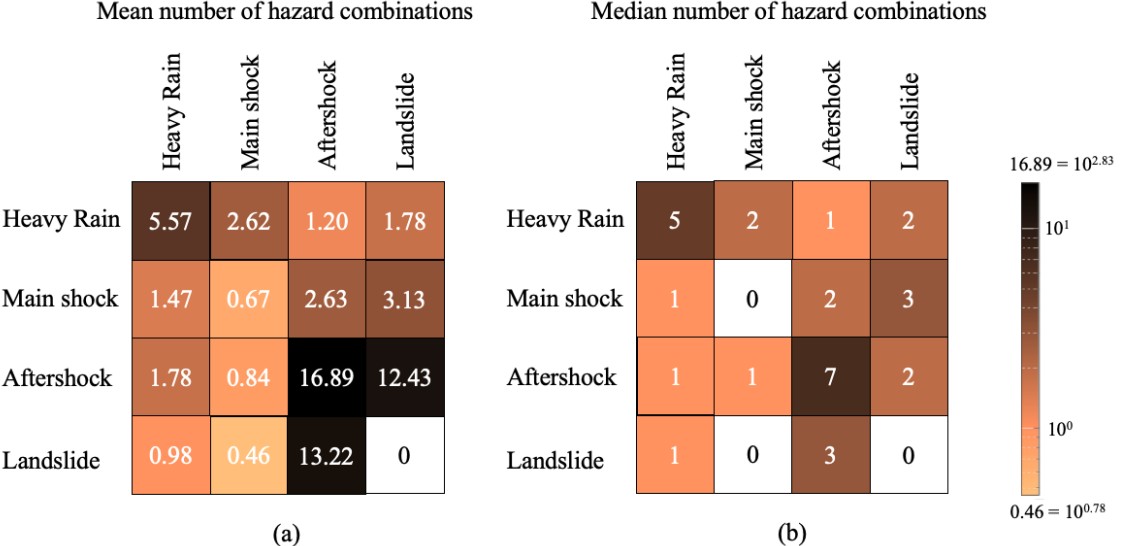

**Figure 9.** Mean (a) and median (b) number of hazard combinations throughout the life cycle hazard event set.

## 5 Conclusions

The paper proposes a simple simulation-based approach to account for different types of hazard interactions in generating a life cycle multi-hazard event set, i.e., a sequence of events throughout a system's life cycle and their associated characteristics/severities. We account for concurrent interactions, successive interactions where the secondary hazard event is immediately triggered by the primary one, and successive interactions where the primary hazard event affects the occurrence rate of the secondary one. Each interaction is incorporated into the simulation differently. Concurrent hazards are modeled based on the rate surface that defines the joint rate of the associated severity measures. Successive triggering interactions are incorporated through the conditional probability of occurrence of the secondary hazard type(s) and the conditional distribution of the associated severity measure. Successive altering interactions are modeled by modifying the secondary hazard type(s) rate curve. The different hazard events are modeled as a set of competing Poisson processes, which may be homogeneous or non-homogeneous. The paper fills a gap in the literature for a quantitative interpretation of multi-hazard occurrences, translating the available, qualitative definitions and classifications into a systematic method to simulate event occurrences. The resulting simulation of one life cycle hazard event set is computationally efficient and can be repeated to obtain relevant statistics of hazard occurrences. By using competing Poisson process and integrating within the same methodology both dependent and independent hazards, the proposed simulation method offers insights not only into the combination of hazards arising from causality (i.e., hazard interactions), but also those emerging from sheer temporal coincidence. The significance of these hazard combinations, especially in the context of their anticipated growth due to climate change effects in the coming years, should not be underestimated. By allowing to modify the rate curves used as input to the model, the proposed algorithm allows to incorporate such climate change aspects. The statistics of hazard occurrences can be used in analytical methods for life cycle

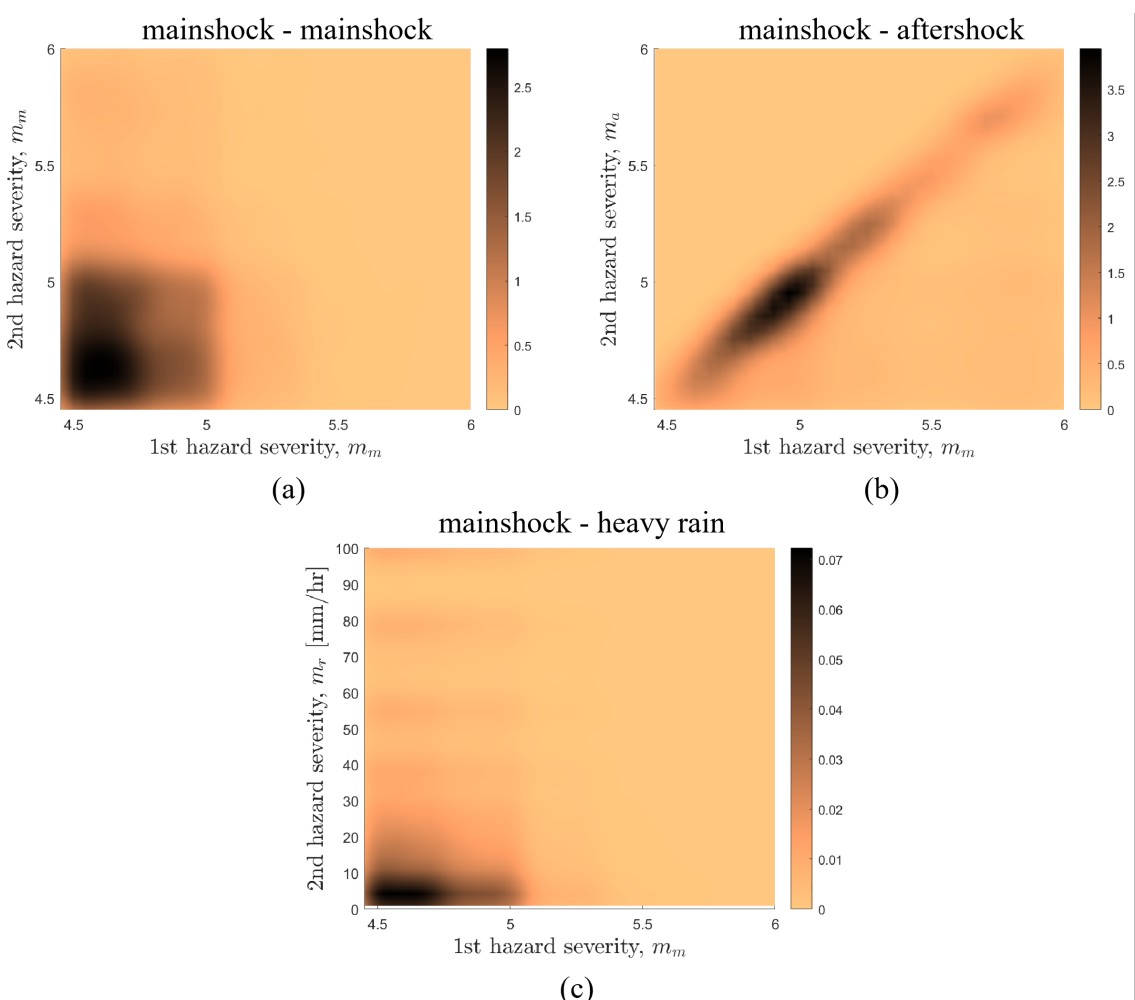

**Figure 10.** Joint probability density function of the severity measures for the hazard combinations mainshock-mainshock (a), mainshock-aftershock (b), and mainshock-heavy rain (c).

analysis to obtain the associated statistics of impact/consequence metrics of interest to end users throughout the service life of a considered system. The simulated event sets can also be integrated into simulation-based frameworks for Level II inter-actions, i.e., impact/consequence interactions that can only occur through the exposed elements. This study specifically delves into the temporal dependence across hazards and does not explicitly discuss any spatial aspects/dependencies. The simulation of the hazard events in the example is based on the rate curves associated with their event characteristics (i.e., location and magnitude for earthquakes or rainfall intensity and duration for heavy rain events). A corresponding local intensity measure for earthquakes (i.e., the peak ground acceleration needed for landslide simulation) is obtained at a single location rather than across a region, although such an extension would be straightforward. In fact, local intensities (for each hazard event) are in general needed for a comprehensive analysis of Level II interactions. For analyses at the regional scale, in particular, the inten-

490 sity measures are in the form of maps (hazard footprints) illustrating the hazard intensity measure's spatial variability across different locations, explicitly accounting for any spatial and cross-intensity correlation (e.g., Jayaram and Baker, 2010a, b).

*Code availability.* The code used to produce the results in this manuscript can be found in the following Github repository: https://github. com/LeandroIannacone/MultiHazardEventSetSimulation. An object-oriented, user-friendly tool to apply the methods presented is currently under development, and it will be available in the following Github repository: https://github.com/robgen/multiHazardEventSet.

*Author contributions.* **Leandro Iannacone:** Writing - Original Draft, Conceptualization, Methodology, Formal Analysis.
**Kenneth Otárola:** Conceptualization, Writing - Review and Editing.
**Roberto Gentile:** Supervision, Conceptualization, Writing - Review and Editing.
**Carmine Galasso:** Supervision, Conceptualization, Writing - Review and Editing.

*Competing interests.* We hereby declare that no competing interest is present.

*Acknowledgements.* The authors acknowledge funding from UKRI GCRF under grant NE/S009000/1, Tomorrow's Cities Hub.

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

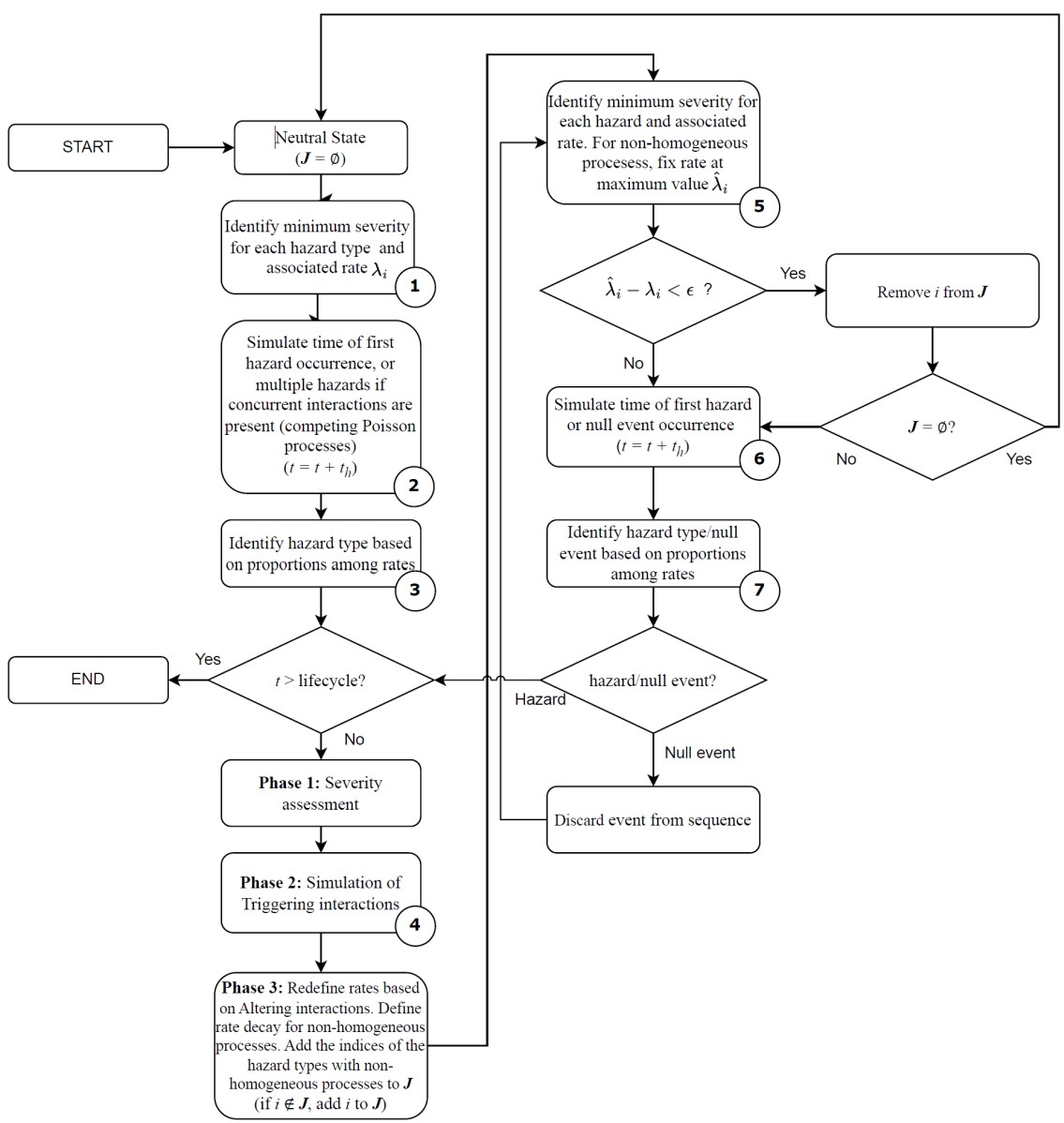

**Figure A1.** Flowchart of the simulation method incorporating time-dependent (non-homogeneous) processes (refer to Section 3.3.2 and Figure 5)

## Appendix A

Below are the flowcharts detailing the sequential Monte Carlo simulation procedure for the cases described in Section 3.3.2 (incorporating time-dependent processes) and Section 3.3.3 (incorporating slow-onset events).

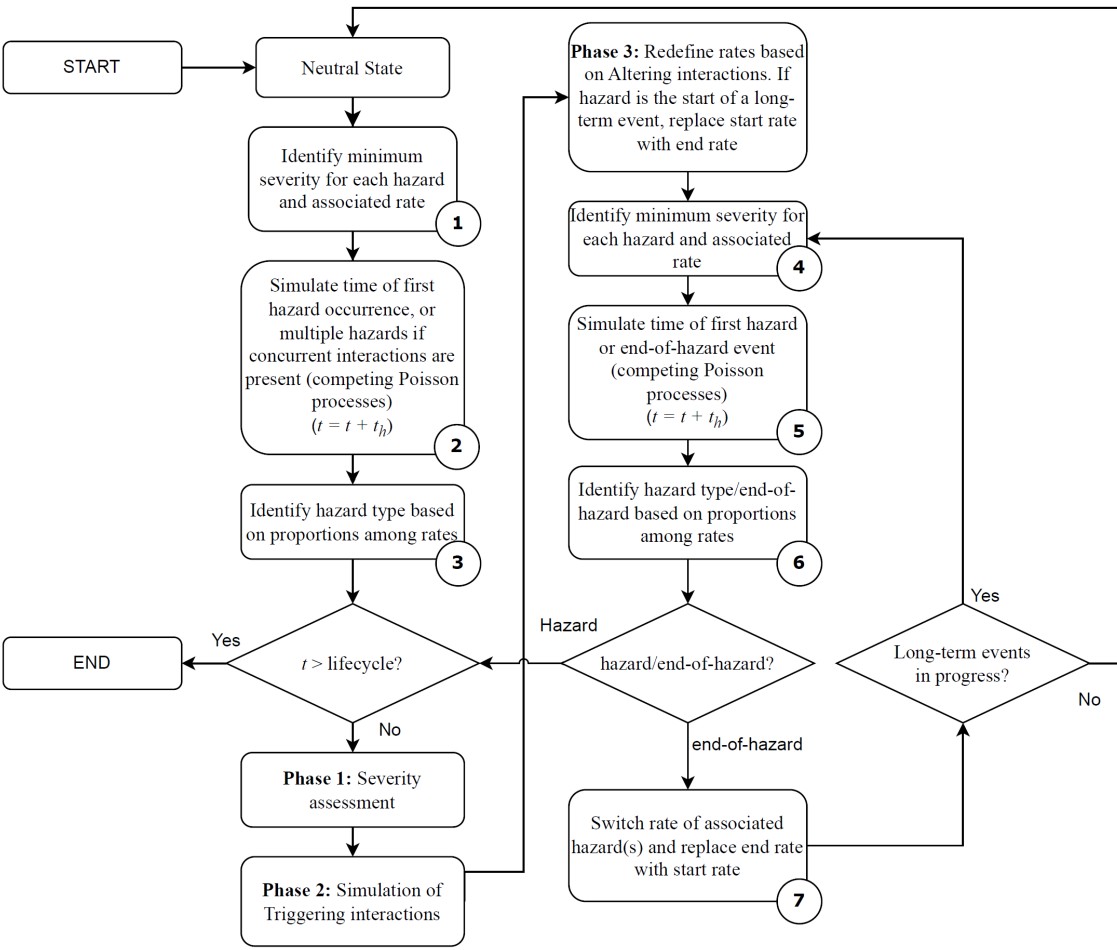

**Figure A2.** Flowchart of the simulation method incorporating slow-onset events (refer to Section 3.3.3 and Figure 6