# Peer review of "Simulating multi-hazard event sets for life cycle consequence analysis"

_EGUsphere, 2023_

## Referee Comment (RC1)

**General Comments:**

The paper *"Simulating Multi-Hazard Event Sets for Life Cycle Consequence Analysis"* by Iannacone et al. is a notable contribution to the field of natural hazard risk quantification and modeling. It innovatively addresses a significant gap in existing literature by proposing a computational framework for simulating sequences of hazard events, considering both Level I (occurrence interactions) and Level II (consequence interactions). The methodological approach, which utilizes competing Poisson processes and a sequential Monte Carlo sampling method, is both rigorous and novel. The paper is well-structured, and the authors provide a comprehensive and clear explanation of their methodology, supported by a numerical example that effectively demonstrates the application and potential of their proposed method. I suggest that the paper be approved for publication after incorporating the modifications and recommendations noted below.

**Specific Comments:**

- The development of a simulation-based method for generating multi-hazard event sets is commendable. The use of competing Poisson processes and a sequential Monte Carlo sampling method to incorporate different types of Level I interactions seems reasonable. However, more clarification on the method choosing and naming other alternative methods to conduct such simulations might be insightful.

- The paper is well-organized, with a logical flow that systematically introduces the problem, the methodology, and a numerical example. Each section builds upon the previous one, making the complex concepts more accessible.

- The inclusion of a detailed numerical example is particularly effective. It not only demonstrates the practical application of the method but also aids in understanding the complexities involved in simulating multi-hazard events. The example is well-chosen and supports the theoretical framework effectively.

- The paper effectively situates the research within the existing scholarly context, highlighting the deficiencies in current methods and the ways in which this study addresses them. Nonetheless, it falls short in providing a thorough literature review, particularly in the introduction, where some cited references are notably outdated. To enhance this aspect, consider including the following recent papers that also explore similar issues among others:

    o Dehghani, N. L., E. Fereshtehnejad, and A. Shafieezadeh. 2021. "A Markovian approach to infrastructure life-cycle analysis: Modeling the interplay of hazard effects and recovery." Earthquake Engng Struct Dyn., 50 (3): 736–755. https://doi.org/10.1002/eqe.3359.
    o Di Baldassarre, G., D. Nohrstedt, J. Mård, S. Burchardt, C. Albin, S. Bondesson, K. Breinl, F. M. Deegan, D. Fuentes, M. G. Lopez, M. Granberg, L. Nyberg, M. R. Nyman, E. Rhodes,

V. Troll, S. Young, C. Walch, and C. F. Parker. 2018. "An Integrative Research Framework to Unravel the Interplay of Natural Hazards and Vulnerabilities." Earth's Future, 6 (3): 305–310. https://doi.org/10.1002/2017EF000764.

o Nofal, O. M., K. Amini, J. E. Padgett, J. W. Van De Lindt, N. Rosenheim, Y. M. Darestani, A. Enderami, E. J. Sutley, S. Hamideh, and L. Duenas-Osorio. 2023. "Multi-hazard socio-physical resilience assessment of hurricane-induced hazards on coastal communities." Resilient Cities and Structures, 2 (2): 67–81. https://doi.org/10.1016/j.rcns.2023.07.003.

o de Ruiter, M. C., A. Couasnon, M. J. C. van den Homberg, J. E. Daniell, J. C. Gill, and P. J. Ward. 2020. "Why We Can No Longer Ignore Consecutive Disasters." Earth's Future, 8 (3): e2019EF001425. https://doi.org/10.1029/2019EF001425.

- The approach has significant implications for risk assessment and disaster management planning. It provides a more realistic assessment of risk by considering the interactions between different hazard types, which is crucial for effective planning and mitigation strategies.

- The combined findings are presented as both mean and median values, as seen in Figure 10, for instance. It would be beneficial to include an explanation of how each value is utilized, as well as clarification of the insights that can be derived from comparing these two measures within the same scenario.

- Enhancing the paper's conclusion with deeper insights into the innovative aspects of the proposed framework, along with more detailed interpretations of potential outcomes and applications, would add value. Additionally, a thorough discussion of the limitations and future research prospects is currently absent and would be a beneficial inclusion.

**Technical Corrections:**

- Figure 2: It would be beneficial to add a label for independent hazards with no interactions in part (a) of the figure, within the legend. Furthermore, the use of arrows in this figure does not effectively aid in understanding the concepts, especially given their inconsistent application. Consider simplifying the representation of interactions in this figure.
- Lines 190-208: The examples of interactions are repetitive, given that definitions and examples have been provided earlier. Please consider eliminating redundant information throughout the paper.
- Figure 5: The parameter '$t_{-1}$' in the figure is unclear, and it lacks a direct mention in the accompanying explanations.
- Line 320: Integrate the sentence about the Appendix into the body of the paragraph, rather than having it as a separate line.
- Equation 13: A parenthesis is missing after '$t_{II}$' in the equation.

- Table 1: The caption is not aligned with the table's position. Instead of a single line mentioning the table, it would be more logical to refer to it before detailing each input in lines 333-334, or even consider placing it in the appendix since all the references are already explained clearly.
- Figure 9: The amount of information in this figure is minimal and could be more effectively presented in a simple table, thereby reducing unnecessary complexity.
- Figure 10: Using the same scale for both parts of the figure would enhance its insightfulness, as it allows for a better visual comparison of the mean and median across the same scenarios. Aim to use a single scale bar for both parts of the figure.

---

## Author Comment (AC1)

**Response to Reviewer 1**

Leandro Iannacone, Kenneth Otárola, Roberto Gentile, Carmine Galasso

January 22, 2024
* * *
**General Comment:** The paper "Simulating Multi-Hazard Event Sets for Life Cycle Consequence Analysis" by Iannacone et al. is a notable contribution to the field of natural hazard risk quantification and modeling. It innovatively addresses a significant gap in existing literature by proposing a computational framework for simulating sequences of hazard events, considering both Level I (occurrence interactions) and Level II (consequence interactions). The methodological approach, which utilizes competing Poisson processes and a sequential Monte Carlo sampling method, is both rigorous and novel. The paper is well-structured, and the authors provide a comprehensive and clear explanation of their methodology, supported by a numerical example that effectively demonstrates the application and potential of their proposed method. I suggest that the paper be approved for publication after incorporating the modifications and recommendations noted below.

*We thank the reviewer for the attentive reading and positive overall feedback. Below, we address each of the reviewer's comments separately.*
* * *
**Specific Comment 1:** The development of a simulation-based method for generating multi-hazard event sets is commendable. The use of competing Poisson processes and a sequential Monte Carlo sampling method to incorporate different types of Level I interactions seems reasonable. However, more clarification on the method choosing and naming other alternative methods to conduct such simulations might be insightful.

*Thank you for the positive feedback and the suggestion. As mentioned in the brief literature review of the manuscript, there are very few alternative methods for dealing quantitatively with the simulation (or statistical analysis) of the occurrence of interacting hazards. We have decided to explicitly include a brief description of the two alternative quantitative approaches we were able to identify, the one by Selva (2013) and the one by Mignan et al. (2014).*

*We plan to add the following discussion on Line 219 (Page 10):*

> To the best of the authors' knowledge, no algorithm is currently available in the literature that accounts for the types of interactions and additional aspects (i.e., event duration and temporal variability) highlighted in this paper. An alternative sequential Monte Carlo approach has been proposed by Mignan et al. (2014), which disaggregates the simulation of primary events from the simulation

of secondary events (all primary events are simulated, then all secondary events are simulated). However, such an algorithm only considers sudden-onset, time-independent hazard events, and models all interactions as successive, triggering. Similarly, Selva (2013) used simplified, closed-form solutions to translate rate curves into probabilities of occurrence of the hazards within a given time period. While the interactions between hazards can be included by modifying the probability of occurrence of the secondary hazard (Selva introduces "co-active risk factors" for this purpose, and provides an example with volcanic eruptions and ash fallout), such an approach is also limited to sudden-onset, time-independent events and does not capture the intricacies of hazard sequences that may include multiple successive interactions.
* * *
**Specific Comments 2-3:** The paper is well-organized, with a logical flow that systematically introduces the problem, the methodology, and a numerical example. Each section builds upon the previous one, making the complex concepts more accessible.

The inclusion of a detailed numerical example is particularly effective. It not only demonstrates the practical application of the method but also aids in understanding the complexities involved in simulating multi-hazard events. The example is well-chosen and supports the theoretical framework effectively.

*We gratefully acknowledge the positive feedback by the reviewer.*
* * *
**Specific Comment 4:** The paper effectively situates the research within the existing scholarly context, highlighting the deficiencies in current methods and the ways in which this study addresses them. Nonetheless, it falls short in providing a thorough literature review, particularly in the introduction, where some cited references are notably outdated. To enhance this aspect, consider including the following recent papers that also explore similar issues among others:

- Dehghani, N. L., E. Fereshtehnejad, and A. Shafieezadeh. 2021. "A Markovian approach to infrastructure life-cycle analysis: Modeling the interplay of hazard effects and recovery." Earthquake Engng Struct Dyn., 50 (3): 736–755. https://doi.org/10.1002/eqe.3359.

- Di Baldassarre, G., D. Nohrstedt, J. Mård, S. Burchardt, C. Albin, S. Bondesson, K. Breinl, F. M. Deegan, D. Fuentes, M. G. Lopez, M. Granberg, L. Nyberg, M. R. Nyman, E. Rhodes, V. Troll, S. Young, C. Walch, and C. F. Parker. 2018. "An Integrative Research Framework to Unravel the Interplay of Natural Hazards and Vulnerabilities." Earth's Future, 6 (3): 305–310. https://doi.org/10.1002/2017EF000764.

- Nofal, O. M., K. Amini, J. E. Padget, J. W. Van De Lindt, N. Rosenheim, Y. M. Darestani, A. Enderami, E. J. Sutley, S. Hamideh, and L. Duenas-Osorio. 2023. "Multi-hazard socio-physical resilience assessment of hurricane-induced hazards on coastal communities." Resilient Cities and Structures, 2 (2): 67–81. https://doi.org/10.1016/j.rcns.2023.07.003.

- de Ruiter, M. C., A. Couasnon, M. J. C. van den Homberg, J. E. Daniell, J. C. Gill, and P. J. Ward. 2020. "Why We Can No Longer Ignore Consecutive Disasters." Earth's Future, 8(3): e2019EF001425. https://doi.org/10.1029/2019EF001425.

*We will expand the literature review to include the references suggested by the reviewer and the following additional references:*

- *Cutter, S. L., Barnes, L., Berry, M., Burton, C., Evans, E., Tate, E., and Webb, J.: A place-based model for understanding community resilience to natural disasters, Global environmental change, 18, 598–606, 2008.*

- *Cutter, S.L.: Compound, cascading, or complex disasters: what's in a name?, Environment: Science and Policy for Sustainable Development, 60(6), 16-25, 2018.*

- *Selva, J.: Long-term multi-risk assessment: statistical treatment of interaction among risks, Natural hazards, 67, 701–722, 2013.*

*We will also included additional paragraphs in the text to contextualize such references.*

*Specifically, we plan to add the following discussion on Line 20 (Page 1):*

> In fact, the occurrence of multiple events within a short time span (whether dictated by a causality between events or by sheer coincidence) may subject the system to exacerbated economic and societal consequences (de Ruiter et al., 2020). Such consequences have been increasing over the past decades (Di Baldassarre et al., 2018) due to several factors such as climate change, urbanization, and globalization (Cutter et al., 2008; Cutter, 2018).

*We plan to add the following discussion on Line 20 (Page 2):*

> The challenges associated with obtaining realistic sequences of events have led multiple authors to select specific, representative scenarios in their multi-hazard assessments, disregarding the Level I interactions in favor of a detailed study of Level II interactions (e.g., Nofal et al., 2023).

*The reference to Dehghani et al. (2021) will be added among the list of studies on Level II interactions.*
* * *
**Specific Comment 5:** The approach has significant implications for risk assessment and disaster management planning. It provides a more realistic assessment of risk by considering the interactions between different hazard types, which is crucial for effective planning and mitigation strategies.

*We gratefully acknowledge the positive feedback by the reviewer.*
* * *
**Specific Comment 6:** The combined findings are presented as both mean and median values, as seen in Figure 10, for instance. It would be beneficial to include an explanation of how each value is utilized, as well as clarification of the insights that can be derived from comparing these two measures within the same scenario.

*The difference between the mean and the median value can be used to effectively assess the skewness of the distribution for the number of events throughout the life cycle. If the mean and median values are close, the distribution does not have many outliers, and the realizations are well-distributed around the mean. If the mean and median values are different, there may be a high variability across realizations. In such cases, there may be multiple event sets that present either a low number of events (even zero events) or an extremely high number of events.*

*We will incorporate such discussion into the manuscript on Line 405 (Page 19), as follows:*

> The difference between the mean and median values can effectively indicate the skewness of the associated distribution of events. When the mean and median are similar, the distribution exhibits few outliers, with realizations evenly distributed around the mean. Conversely, a notable difference between the mean and median signals heightened variability among realizations, suggesting the presence of multiple event sets with either a scarcity (e.g., zero or close to zero) or a high number of events.
* * *
**Specific Comment 7:** Enhancing the paper's conclusion with deeper insights into the innovative aspects of the proposed framework, along with more detailed interpretations of potential outcomes and applications, would add value. Additionally, a thorough discussion of the limitations and future research prospects is currently absent and would be a beneficial inclusion.

*Following the reviewer's suggestion, we will expand the conclusions section to incorporate a more detailed discussion on the novelties of the manuscript, its limitations, and future research prospects.*

*We will add the following discussion on Line 435 (Page 21):*

> The paper fills a gap in the literature for quantitative modeling of multi-hazard occurrences, translating the available qualitative definitions and classifications into a systematic method to simulate event occurrences.

*We will add the following discussion on Line 436 (Page 22):*

By using competing Poisson processes and integrating both dependent and independent hazards within the same methodology, the proposed simulation approach offers insights into the combination of hazards arising from causality (i.e., hazard interactions) and those emerging from sheer temporal coincidence. The significance of these hazard combinations, especially in the context of their anticipated growth due to climate change effects in the coming years, should not be underestimated. By allowing the modification of the rate curves used as input to the model, the proposed algorithm enables incorporating such climate change aspects.

*We will add the following discussion on Line 439 (Page 22):*

This study specifically delves into the temporal dependence across hazards and does not explicitly discuss any spatial aspects/dependencies. The simulation of the hazard events in the example is based on the rate curves associated with their event characteristics (i.e., location and magnitude for earthquakes or rainfall intensity and duration for heavy rain events). A corresponding local intensity measure for earthquakes (i.e., the peak ground acceleration needed for landslide simulation) is obtained at a single location rather than across a region, although such an extension would be straightforward. Local intensities (for each hazard event) are in general needed for a comprehensive analysis of Level II interactions. For analyses at the regional scale, the intensity measures are in the form of maps (hazard footprints) illustrating the hazard intensity measure's spatial variability across different locations, explicitly accounting for any spatial and cross-intensity correlation (e.g., Jayaram and Baker, 2010a,b).
* * *
**Technical Correction 1:** Figure 2: It would be beneficial to add a label for independent hazards with no interactions in part (a) of the figure, within the legend. Furthermore, the use of arrows in this figure does not effectively aid in understanding the concepts, especially given their inconsistent application. Consider simplifying the representation of interactions in this figure.

*We will update Figure 2 to incorporate the reviewer's comment. The revised figure is reported below.*

[Figure]

*Figure 7 will also be updated for consistency.*
* * *
**Technical Correction 2:** Lines 190-208: The examples of interactions are repetitive, given that definitions and examples have been provided earlier. Please consider eliminating redundant information throughout the paper.

*We thank the reviewer for the comment. However, after careful thought, we plan to leave these lines unchanged. The examples in the previous sections are meant as qualitative introductions to the types of interactions that can exist between hazards. The examples in this section, however, are meant to point the reader toward resources of mathematical models to simulate those interactions. Specifically, we refer to the work from Neri et al. (2008), which contains examples for both the conditional probabilities and distributions mentioned earlier. We will slightly modify the text in this section to make sure it is clear why these*

*examples are presented. We will also carefully review the entirety of the manuscript to eliminate redundant information.*

*Line 192 (Page 9) will be modified to read as follows:*

> An example of conditional probabilities and conditional distributions used to model triggering interactions can be found in Neri et al. (2008)...
* * *
**Technical Correction 3:** Figure 5: The parameter '$t_{-1}$' in the figure is unclear, and it lacks a direct mention in the accompanying explanations.

*We thank the reviewer for the attentive reading. This is indeed a typo coming from a superseded version of the mathematical formulation. We will replace the parameter $\lambda_3(t_{-1})$ with $\hat{\lambda}_3$, which is the notation used in the text.*
* * *
**Technical Correction 4:** Line 320: Integrate the sentence about the Appendix into the body of the paragraph, rather than having it as a separate line.

*This is actually an issue with the automatic formatting of the LaTeX document. The sentence is actually placed after Figure 6, but it was moved to the current location for layout purposes. We will ensure the issue does not appear in the final, proofed version of the manuscript.*
* * *
**Technical Correction 5:** Equation 13: A parenthesis is missing after $'t'_{ll}$ in the equation.

*We will fix the typo. Below is the revised equation.*

$$s_l(t) = \begin{cases} 1 & \text{if } t < t_{l1} \\ 15e^{-0.12(t-t_{ll})} & \text{if } t > t_{l1} \end{cases}$$
* * *
**Technical Correction 6:** Table 1: The caption is not aligned with the table's position. Instead of a single line mentioning the table, it would be more logical to refer to it before detailing each input in lines 333-334, or even consider placing it in the appendix since all the references are already explained clearly.

*We will increase the table's width so that it spans the full page width and is aligned with the caption. Following the reviewer's suggestion, we will also move the table earlier in the manuscript. It will appear after Line 334 on Page 15.*
* * *
**Technical Correction 7:** Figure 9: The amount of information in this figure is minimal and could be more effectively presented in a simple table, thereby reducing unnecessary complexity.

*We agree with the reviewer and we will transform the former Figure 9 into a table (Table 2). We report the table below.*

| Hazard type | Mean number of events | Median number of events |
|---|---|---|
| Heavy Rain | 25.86 | 26 |
| Mainshock | 12.02 | 12 |
| Aftershock | 36.02 | 17.5 |
| Landslide | 17.36 | 8 |
* * *
**Technical Correction 8:** Figure 10: Using the same scale for both parts of the figure would enhance its insightfulness, as it allows for a better visual comparison of the mean and median across the same scenarios. Aim to use a single scale bar for both parts of the figure.

*We will revise Figure 10 to use a single scale for both parts of the figure. The new figure is reported below.*

[Figure]

**References**

Cutter, S. L.: Compound, cascading, or complex disasters: what's in a name?, Environment: Science and Policy for Sustainable Development, 60, 16–25, 2018.

Cutter, S. L., Barnes, L., Berry, M., Burton, C., Evans, E., Tate, E., and Webb, J.: A place-based model for understanding community resilience to natural disasters, Global environmental change, 18, 598–606, 2008.

de Ruiter, M. C., Couasnon, A., van den Homberg, M. J., Daniell, J. E., Gill, J. C., and Ward, P. J.: Why we can no longer ignore consecutive disasters, Earth's future, 8, e2019EF001 425, 2020.

Dehghani, N. L., Fereshtehnejad, E., and Shafieezadeh, A.: A Markovian approach to infrastructure life-cycle analysis: Modeling the interplay of hazard effects and recovery, Earthquake Engineering & Structural Dynamics, 50, 736–755, 2021.

Di Baldassarre, G., Nohrstedt, D., Mård, J., Burchardt, S., Albin, C., Bondesson, S., Breinl, K., Deegan, F. M., Fuentes, D., Lopez, M. G., et al.: An integrative research framework to unravel the interplay of natural hazards and vulnerabilities, Earth's Future, 6, 305–310, 2018.

Jayaram, N. and Baker, J. W.: Considering spatial correlation in mixed-effects regression and the impact on ground-motion models, Bulletin of the Seismological Society of America, 100, 3295–3303, 2010a.

Jayaram, N. and Baker, J. W.: Efficient sampling and data reduction techniques for probabilistic seismic lifeline risk assessment, Earthquake Engineering & Structural Dynamics, 39, 1109–1131, 2010b.

Mignan, A., Wiemer, S., and Giardini, D.: The quantification of low-probability–high-consequences events: part I. A generic multi-risk approach, Natural Hazards, 73, 1999–2022, 2014.

Neri, A., Aspinall, W. P., Cioni, R., Bertagnini, A., Baxter, P. J., Zuccaro, G., Andronico, D., Barsotti, S., Cole, P. D., Ongaro, T. E., et al.: Developing an event tree for probabilistic hazard and risk assessment at Vesuvius, Journal of volcanology and geothermal research, 178, 397–415, 2008.

Nofal, O. M., Amini, K., Padgett, J. E., van de Lindt, J. W., Rosenheim, N., Darestani, Y. M., Enderami, A., Sutley, E. J., Hamideh, S., and Duenas-Osorio, L.: Multi-hazard socio-physical resilience assessment of hurricane-induced hazards on coastal communities, Resilient Cities and Structures, 2, 67–81, 2023.

Selva, J.: Long-term multi-risk assessment: statistical treatment of interaction among risks, Natural hazards, 67, 701–722, 2013.

---

## Author Comment (AC2)

**Response to Reviewer 2**

Leandro Iannacone, Kenneth Otárola, Roberto Gentile, Carmine Galasso

January 22, 2024
* * *
**General Comment:** This is an ambitious paper simulating multi-hazard event sets for life cycle consequence analysis. The methodology is transparent and is systematically presented, and should meet the general requirements of life cycle consequence analysis. However, it should be made clear that uncommon combinations of different hazards can create dangerous hazard situations. With progressive climate change, these dangerous hazard situations may arise more often. Thus the occurrence of a typhoon close to an earthquake can have a major impact on fire and landslide risk, as demonstrated in Japan in September 1923 in Tokyo, and September 2018 in Hokkaido.

Some systems may demand a very high degree of life cycle reliability. The authors should stress test their Poisson process modelling to gauge the sensitivity of the model results to anomalous rare compound event behaviour. End users should not be surprised by such phenomena.

*We thank the reviewer for the attentive reading of the manuscript, and we hereby gratefully acknowledge the positive remarks. Indeed, the formulation presented in this paper can capture (and quantify the probability of) the occurrence of rare combinations of events that arise from sheer temporal coincidence rather than causality (i.e., interactions). Employing competing Poisson processes in the formulation offers a distinct advantage in accommodating both types of phenomena. This allows the model not only to identify commonly considered sequences of dependent events, such as mainshocks and aftershocks or heavy rain and flooding but also to enumerate 'coincidental' combinations of hazards. For instance, scenarios like a typhoon occurring in close (temporal and spatial) proximity to an earthquake, as highlighted by the reviewer, can be systematically incorporated into the analysis. We will modify and add several sentences in the manuscript to stress this aspect.*

*We plan to add the following discussion on Line 20 (Page 1):*

In fact, the occurrence of multiple events within a short time span (whether dictated by a causality between events or by sheer coincidence) may subject the system to exacerbated economic and societal consequences (de Ruiter et al., 2020). Such consequences have been increasing over the past decades (Di Baldassarre et al., 2018) due to several factors such as climate change, urbanization, and globalization (Cutter et al., 2008; Cutter, 2018).

*We plan to add the following discussion on Line 417 (Page 20):*

> This sort of 'coincidental' hazard combinations should not surprise end users of the algorithm. In fact, such combinations have been observed on multiple occasions over the past century and are expected to increase due to climate change (Cutter et al., 2008). For example, typhoons were recorded in close temporal proximity to the great Kanto earthquake (Japan) of 1923 (Sasaki and Yamakawa, 2007) and the Hokkaido earthquake (Japan) of 2018 (Heidarzadeh et al., 2023). Although less frequent, these combinations are just as crucial as those influenced by causality (de Ruiter et al., 2020). An additional advantage of the proposed method is that it seamlessly integrates causal and coincidental event combinations within the same formulation.

*We plan to add the following discussion on Line 436 (Page 22):*

> By using competing Poisson processes and integrating both dependent and independent hazards within the same methodology, the proposed simulation approach offers insights into the combination of hazards arising from causality (i.e., hazard interactions) and those emerging from sheer temporal coincidence. The significance of these hazard combinations, especially in the context of their anticipated growth due to climate change effects in the coming years, should not be underestimated. By allowing the modification of the rate curves used as input to the model, the proposed algorithm enables incorporating such climate change aspects.

*We believe that the numerical example provided in the manuscript already showcases the method's potential to capture the presence of rare, coincidental hazard combinations.*

*For example, Lines 413-417 on Page 21 read:*

> It can be observed that even hazard types that are not related by level I interactions may occur close (in time) to each other. For example, there are, on average 2.62 main shock events following the occurrence of a heavy rain event and 1.47 heavy rain events following a mainshock event, which might suggest that the interactions between such (independent) hazards might be of interest in a possible life cycle analysis of a structure placed in the investigated location.

*As for the consequences of such rare hazard combinations, their investigation is deferred to works on Level II interactions, which are not the focus of this paper, but are explored in other works by the authors such as Otárola et al. (2023a,b) .*

**General Comment 2:** The authors should address

*We interpreted this sentence as a typo in the review, and we have no further comments. In case the review was somehow submitted incompletely, we will be happy to address any of the reviewer's additional comments.*

**References**

Cutter, S. L.: Compound, cascading, or complex disasters: what's in a name?, Environment: Science and Policy for Sustainable Development, 60, 16–25, 2018.

Cutter, S. L., Barnes, L., Berry, M., Burton, C., Evans, E., Tate, E., and Webb, J.: A place-based model for understanding community resilience to natural disasters, Global environmental change, 18, 598–606, 2008.

de Ruiter, M. C., Couasnon, A., van den Homberg, M. J., Daniell, J. E., Gill, J. C., and Ward, P. J.: Why we can no longer ignore consecutive disasters, Earth's future, 8, e2019EF001 425, 2020.

Di Baldassarre, G., Nohrstedt, D., Mård, J., Burchardt, S., Albin, C., Bondesson, S., Breinl, K., Deegan, F. M., Fuentes, D., Lopez, M. G., et al.: An integrative research framework to unravel the interplay of natural hazards and vulnerabilities, Earth's Future, 6, 305–310, 2018.

Heidarzadeh, M., Miyazaki, H., Ishibe, T., Takagi, H., and Sabeti, R.: Field surveys of September 2018 landslide-generated waves in the Apporo dam reservoir, Japan: combined hazard from the concurrent occurrences of a typhoon and an earthquake, Landslides, 20, 143–156, 2023.

Otárola, K., Iannacone, L., Gentile, R., and Galasso, C.: Multi-hazard life-cycle consequence analysis of deteriorating engineering systems, Structural Safety (under review), 2023a.

Otárola, K., Iannacone, L., Gentile, R., and Galasso, C.: A Markovian framework for multi-hazard life-cycle consequence analysis of deteriorating structural systems, in: Proceedings of the 14th International Conference on Applications of Statistics and Probability in Civil Engineering, ICASP14, Dublin, Ireland, pp. 9–13, 2023b.

Sasaki, H. and Yamakawa, S.: Natural hazards in Japan, in: International perspectives on natural disasters: occurrence, mitigation, and consequences, pp. 163–180, Springer, 2007.